# CellSighter: a neural network to classify cells in highly multiplexed images

Yael Amitay[1,2], Yuval Bussi[1,2], Ben Feinstein[2], Shai Bagon[2], Idan Milo[1] & Leeat Keren[1] ✉

Multiplexed imaging enables measurement of multiple proteins in situ, offering an unprecedented opportunity to chart various cell types and states in tissues. However, cell classification, the task of identifying the type of individual cells, remains challenging, labor-intensive, and limiting to throughput. Here, we present CellSighter, a deep-learning based pipeline to accelerate cell classification in multiplexed images. Given a small training set of expert-labeled images, CellSighter outputs the label probabilities for all cells in new images. CellSighter achieves over 80% accuracy for major cell types across imaging platforms, which approaches inter-observer concordance. Ablation studies and simulations show that CellSighter is able to generalize its training data and learn features of protein expression levels, as well as spatial features such as subcellular expression patterns. CellSighter's design reduces overfitting, and it can be trained with only thousands or even hundreds of labeled examples. CellSighter also outputs a prediction confidence, allowing downstream experts control over the results. Altogether, CellSighter drastically reduces hands-on time for cell classification in multiplexed images, while improving accuracy and consistency across datasets.

The spatial organization of tissues facilitates healthy function and its disruption contributes to disease[1]. Recently, a suite of multiplexed imaging technologies has been developed, which enable measurement of the expression of dozens of proteins in tissue specimens at single-cell resolution while preserving tissue architecture[2–14]. These technologies open new avenues for large-scale molecular analysis of human development, health and disease. However, while technologies have developed rapidly, with datasets spanning thousands of images[15,16], data analysis presents a major limitation to throughput. Specifically, cell classification, the task of identifying different cell types in the tissue remains an inaccurate, slow and laborious process.

Analysis of multiplexed images has converged on a common sequence of procedures (Fig. 1A). While technologies differ in implementation, from cyclic fluorescence to mass-spectrometry, they all generate a stack of images, each depicting the expression of one protein in the tissue. Initial processing corrects technology-specific artifacts such as autofluorescence, noise, and image registration[17–19].

Next, images undergo *cell segmentation* to identify individual cells in the tissue. Recently, artificial intelligence (AI) algorithms, trained on large manually-curated datasets, have automated this task, approaching human-level performance[20–22]. Next, the expression of each protein is quantified in each cell to create an *expression matrix*. This table serves as input for *cell classification*, where the type and phenotype of each cell is inferred from co-expressed proteins, in combination with prior knowledge. For example, a cell expressing CD45 will be classified as an immune cell. A cell that, in addition, expresses CD3 and CD8 is a cytotoxic T cell, and if that cell also expresses PD-1, LAG-3, and TIM-3, it is classified as an exhausted cytotoxic T cell[23].

Cell classification methods typically involve manual gating or clustering of the expression matrix using algorithms that were developed for isolated cells, such as cytometry or single cell RNA sequencing (scRNAseq)[8,13,19,24–30]. However, deriving cell classifications from multiplexed images has unique challenges over classifying cells in suspension, due to biological and technical factors. For example,

---

[1]Department of Molecular Cell Biology, Weizmann Institute of Science, Rehovot, Israel. [2]Department of Mathematics and Computer Science, Weizmann Institute of Science, Rehovot, Israel. ✉e-mail: leeat.keren@weizmann.ac.il

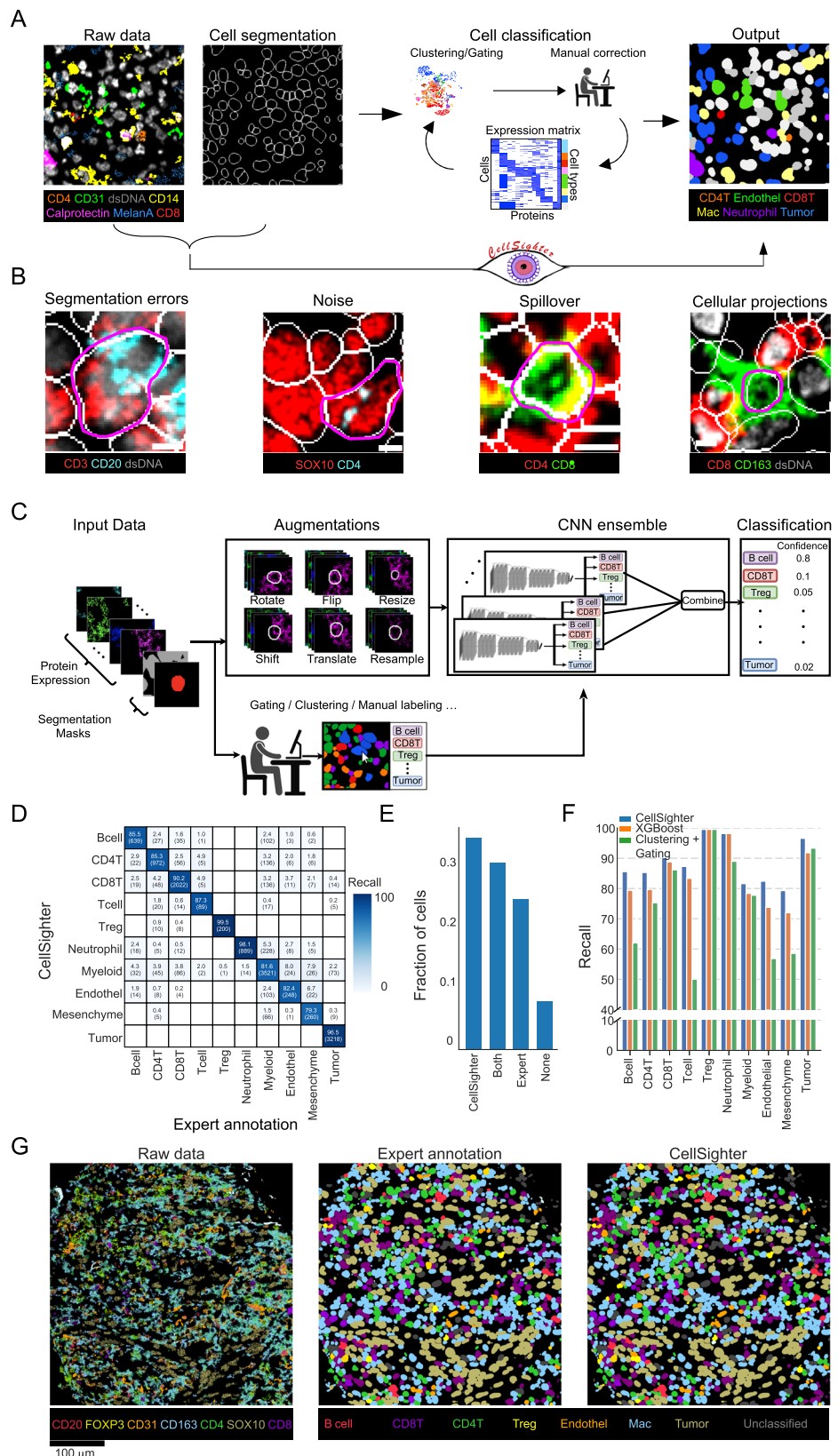

Fig. 1B (I) shows an example of a B cell and a T cell which were erroneously segmented as one cell. In the expression matrix, this cell will appear as expressing both CD20 and CD3. Imaging artifacts also make classification challenging[17]. Figure 1B (II) shows an example of a tumor cell, with overlapping noise in CD4. In the expression matrix, this cell will erroneously appear to express CD4.

Biological factors also contribute to the difficulty of cell classification from images. In tissues, cells form densely-packed communities, as shown for the cytotoxic T cell closely interacting with T helper cells in Fig. 1B (III). Moreover, cells extend projections to facilitate trans-cellular interactions[31], as shown for the CD163+ macrophage in Fig. 1B (IV). This close-network of cell bodies and projections results

**Fig. 1 | CellSighter–a convolutional neural network for cell classification.**
**A** Standard pipelines for cell classification take in multiplexed images and cell segmentation masks and generate an expression matrix. Cells in the matrix are annotated by rounds of clustering, gating, visual inspection and manual correction. CellSighter works directly on the images. **B** Imaging artifacts and biological factors contribute to making cell classification from images challenging. Segmentation errors, noise, tightly packed cells and cellular projections are easily visible in images, but hard to discern in the expression matrix. Scale bar = 5 μm. **C** CellSighter is an ensemble of convolutional neural networks (CNNs) to perform supervised classification of cells. **D** Comparison between labels generated by experts (x-axis) and labels generated by CellSighter (y-axis) shows good agreement. **E** Expert inspection of additional cells differing in classification between CellSighter and expert annotation. Inspection was performed blindly, without knowledge of the source of the label. **F** Comparing the recall of CellSighter (blue), XGBoost trained on the same labels as CellSighter (orange) and Clustering and gating (green). **G** For one field of view (FOV), shown are the protein expression levels (left), expert-generated labels (middle) and CellSighter labels (right). Source data are provided as a Source Data file.

in *spillover*, whereby the protein signals from one cell overlap with the pixels of nearby cells. Several works have proposed methods to deal with spillover using compensation[3,32], pixel analysis[33] or neighborhood analysis[34], but these suffer from signal attenuation, difficulty in scaling to large datasets, or requirements for additional data sources such as scRNAseq on the same tissue. Altogether, cell classification has hitherto remained a time-consuming and labor-intensive task, requiring investigators to perform sequential rounds of clustering, gating, visual inspection and manual annotation (Fig. 1A and Supplementary Fig. 1A). Accordingly, the accuracy of classification is often user-dependent and may impede the quality of downstream analysis.

In this work, we sought to accelerate and improve cell classification from multiplexed imaging by harnessing two insights into this task. First, the effects of segmentation errors, noise, spillover and projections accumulate over the millions of cells routinely measured in multiplexed imaging datasets. As a result, the time that it takes to classify cells using sequential rounds of clustering, gating, image inspection and manual labeling is proportional to the number of cells in the dataset. It is easier and faster to generate curated classifications for thousands of cells than for millions of cells. This observation implies that a machine-learning approach that learns classifications from a subset of the data and transfers them to the rest of the dataset could largely expedite the process of cell classification as suggested[35]. Second, while segmentation errors, noise, spillover and projections confound protein expression values in the expression matrix, they are often distinguishable in images (Supplementary Fig. 1A). We therefore reasoned that a computer-vision approach that works directly on the images as input, rather than on the expression matrix, may have better performance in the task of cell classification. Specifically, deep convolutional neural networks (CNNs) have had remarkable success in computer vision tasks and have recently gained impact in medical imaging, from radiology to electron microscopy[20,22,36,37].

Here, we present CellSighter, a deep-learning based pipeline to perform cell classification in multiplexed images. Given multiplexed images, segmentation masks and a small training set of expert-labeled images, CellSighter outputs the probability of each cell to belong to different cell types. We tested CellSighter on data from different multiplexed imaging modalities and found that it achieves 80–100% recall for major cell types, which approaches inter-observer concordance. Ablation studies and simulations showed that CellSighter learns features of protein expression levels, but also spatial features such as subcellular expression patterns and spillover from neighboring cells. Cell-Sighter's design reduces overfitting and it can be easily trained on only thousands or even hundreds of labeled examples, depending on cell type. Importantly, CellSighter also outputs confidence in prediction, allowing an expert to evaluate the quality of the classifications and tailor the prediction accuracy to their needs. Finally, CellSighter can be applied across datasets, which facilitates cross-study data integration and standardization. Altogether, CellSighter drastically reduces hands-on time for cell classification in multiplexed images, while improving accuracy and consistency across datasets.

## Results

### CellSighter–a convolutional neural network for cell classification

We designed CellSighter as an ensemble of CNN models, each performing multi-class classification. Given raw multiplexed images, as well as the corresponding segmentation mask, CellSighter returns the probability of each cell to belong to one of several classes (Fig. 1C). The input for each model is a 3-dimensional tensor, consisting of cropped images of K proteins centered on the cell to be classified (Supplementary Fig. 1B, C). To incorporate the information of the segmentation, but in a soft manner, we added two additional images to the tensor. The first consists of a binary mask for the cell we want to classify (1's inside the cell and 0's outside the cell), and the second is a similar binary mask for all the other cells in the environment. To deal with large class imbalances between cell types, which in tissues can easily reach 100-fold[26,27], when training the network, we upsampled rare cells such that the major lineages are represented in equal proportions (Methods). However, for very low-abundant classes upsampling alone can result in spurious correlations and overfitting. We, therefore, added standard and custom augmentations to the data, including rotations and flips, minor resizing of the segmentation mask, minor shifts of the images relative to each other, and signal averaging followed by Poisson sampling (Methods).

The final cell classification is given by integrating the results from ten separately-trained CNNs. Each model predicts the probability of all classes. We then average those probabilities and take the class with the maximum probability as the final prediction and the probability value as the confidence. We found that this design provides results that are more robust and reduces grave errors of the network, including hallucinations[38]. The confidence score gives the investigators of the data freedom to further process CellSighter's results to decide on the level of specificity, sensitivity and coverage of cells in the dataset that are best for their specific needs. Low-confidence cells can also be used to guide and refine further labeling.

We first tested CellSighter on a dataset of human melanoma metastases, acquired by MIBI-TOF[2]. We took 111 0.5 × 0.5 mm² images, encompassing 145,668 cells and generated highly-curated ground-truth labels for all cells using gating and sequential rounds of visual inspection and manual annotation (Methods). Altogether, we distinguished ten cell types, including different types of immune cells, tumor cells, stromal cells and vasculature.

We trained CellSighter on 101 images and tested it on 10 held-out images. Training the network took one hour for one model in the ensemble and prediction took a few minutes (see Methods for further details on run times). Prediction recall on the test was high (88 ± 7%) (Fig. 1D), and equivalent to the concordance between two different human labelers (86 ± 17%, Supplementary Fig. 1D), suggesting labeler-specific biases in annotation. It ranged from 99% on easily-distinguishable cells, which were defined by nuclear proteins, such as tumor cells (expressing SOX10) and T regulatory cells (Tregs, expressing FoxP3) to ~80% on rare, entangled or lineage-related cell types, mostly defined by membrane proteins such as myeloid and mesenchymal cells. We evaluated to what extent these confusions represent errors in CellSighter, errors in the expert annotations or ambiguous cells. To this end, we assigned another expert to manually

inspect additional 246 random cells that received different annotations by CellSighter and the expert, without knowing which approach provided which annotation. We found that in 31% of cases the confused cells were ambiguous and could be classified as either type, in 8% they were both wrong, in 35% manual inspection agreed with CellSighter and in 25% with the expert (Fig. 1E). Overall, this suggests that discrepancies are mostly driven by ambiguous cells, and CellSighter performs comparably to human labelers. Moreover, in 29% of cases where the expert was correct, the correct classification was the second ranking option (Supplementary Fig. 1E), and correct predictions received higher overall confidence (Supplementary Fig. 1F).

To further benchmark CellSighter's performance, we compared it to two commonly-used approaches. First, we compared it to clustering. We used FlowSOM[24] to cluster the cells to 100 clusters, and then annotated the clusters based on the expression matrix. Second, we used the same data that was used to train CellSighter to train a gradient boosting classifier (XGBoost) that works on the expression matrix[39]. Evaluation on the test showed that CellSighter outperformed Clustering, with a recall of $88.6 \pm 7$ compared to $75 \pm 16$. CellSighter also outperformed XGBoost in eight out of ten categories and had the same performance on the remaining two categories, with an average increase in recall of 4% ($88.6 \pm 7\%$ compared with $84.5 \pm 9\%$, Fig. 1F, Supplementary Fig. 1G). Altogether, visual inspection of the test images confirmed that CellSighter indeed recapitulated both the predictions of individual cells and the large-scale tissue organization (Fig. 1G).

## CellSighter learns protein coexpression patterns

We explored which features drive CellSighter's predictions. First, we checked whether a CNN running on a tensor of protein images was able to learn protein expression per cell type, similar to what an expert does when working on the expression matrix. To perform this analysis, we turned to a second dataset of melanoma metastases in lymph nodes. Lymph nodes are particularly challenging for cell type classification, due to the high density of cells in the tissue. We generated initial labels for all cells using established approaches, including FlowSOM clustering[24], pixel clustering[33], gating and sequential rounds of visual inspection and manual annotation, altogether annotating 116,808 cells from sixteen images into fourteen cell types (Methods). We trained CellSighter on twelve images, and predicted the labels for an additional four, resulting in high overall recall of $85 \pm 8$ (Supplementary Fig. 2A).

We correlated between the cellular protein expression levels and CellSighter's confidence in prediction. For example, the confidence in predicting Neutrophil was highly correlated with the cellular expression levels of Calprotectin ($R = 0.76$, $P < 10^{-20}$, Fig. 2A). Performing this analysis for all proteins across all cell types revealed expected associations between cell types and their respective proteins (Fig. 2B). For example, the confidence of B cell classification was mostly positively correlated with the expression of CD20 ($R = 0.68$, $P = P < 10^{-20}$) and to a lesser extent with CD45RA ($R = 0.5$, $P < 10^{-20}$) and CD45 ($R = 0.28$, $P = P < 10^{-20}$). We also found that CellSighter is aware of the problems of spillover and multi-class classification as B cell classification was also mildly negatively correlated with the expression of CD3 ($R = -0.25$, $P < 10^{-20}$), CD4 ($R = -0.23$, $P < 10^{-20}$) and CD8 ($R = -0.19$, $P < 10^{-20}$). Indeed, a scatter plot of cellular CD20 expression (a hallmark protein for B cells) versus cellular CD8 expression (a hallmark protein for cytotoxic T cells) revealed that CellSighter was confident in its classifications for cells that had high expression of one of these proteins, but had lower confidence in the classification of cells that expressed both proteins (Fig. 2C).

To further probe CellSighter's classification process, we examined the gradients of the network using guided back propagation[40,41]. This analysis provides a value for each pixel in each channel, which indicates how strongly it influenced cell classification (Methods). We found that the gradients were concentrated in the center cell (Fig. 2D) and that they match the expected protein expression patterns. For

example, Fig. 2D shows an example of a cell classified as a Treg, where prominent gradients are observed for FoxP3 inside the cell, but not in the neighboring cells. Weaker gradients are observed for CD4, but not CD45, reflecting their respective roles in classifying Tregs. In cells classified as tumor cells, the strongest gradients were traced back to the images of MelanA and SOX10, whereas for HEVs it was in the images of CD31 and PNAd (Fig. 2E). Altogether, these analyses suggest that protein expression levels are a major determinant of CellSighter's classification process, similar to gating and clustering.

## CellSighter learns spatial expression features

Next, we examined whether CellSighter was able to leverage the fact that it works directly on the images and learn spatial features to aid in classification. Since CellSighter was trained on data resulting from clustering the expression matrix, which suffers from spillover, we wondered whether it could generalize to learn spatial expression patterns. To do this we employed three complimentary approaches: contrasting CellSighter with a machine learning model that works on the expression matrix rather than on the images, analyzing performance on simulated data, and examining the network's gradients.

First, we evaluated whether CellSighter was more robust to spillover compared with using machine learning approaches that work on the expression matrix, such as XGBoost. Visual inspection of the images suggested that CellSighter was more robust to spillover. For example, inspection of Calprotectin showed that for each patch of signal CellSighter tended to classify less cells overlapping with that patch as Neutrophils (Fig. 3A and Supplementary Fig. 2B). Moreover, cells that were classified as neutrophils by CellSighter were mostly the cells that had higher overlap with the signal (>20%, Fig. 3B). To verify that this observation was causal, we performed a simulation where we took a patch of Calprotectin signal and moved it to vary its overall overlap with the cell (Supplementary Fig. 2C, D). We found that for low overlap (<20%) CellSighter was 9% less likely to classify cells as Neutrophils compared to the XGBoost, whereas for higher degrees of overlap (>30%) this trend flipped (Supplementary Fig. 2C, D).

To further examine CellSighter's ability to learn spatial expression patterns, we performed a direct simulation of spillover that generates contradicting expression patterns. Here, we took crops centered on T helper cells and removed their cognate CD4 and CD20 signals, to avoid any confounding factors incurred by the original signals. We then reintroduced CD4 as a membranous signal, and a patch of strong, partially-overlapping CD20 signal, to simulate spillover (Fig. 3C). To verify that our simulation is relevant to real-world data, the expression levels for both CD4 and CD20 were compatible with the distribution of observed values in the dataset (Supplementary Fig. 2E). We then ran both CellSighter on the images and XGBoost on the expression matrix of the resulting cells. Not surprisingly, if we only added the CD4 signal, both models classified 70–80% of cells as CD4 T cells (Supplementary Fig. 2F). However, when introducing the CD20 signal, CellSighter classified 30% of the cells as B cells, whereas XGBoost classified 47% as B cells (Fig. 3D). Moreover, CellSighter had overall lower confidence in these classifications than XGBoost. For example, CellSighter had high confidence (>0.9) that 1.5% of these simulated cells are B cells, compared to 30% for XGBoost (Fig. 3E). Altogether, we conclude that both in real data and in simulations, running a CNN on images is more robust to spillover.

Next, we evaluated whether CellSighter was able to learn the subcellular expression patterns of different proteins. Visual inspection of the gradient maps for several cells suggested that the gradients of nuclear proteins were concentrated in the center of the cell, whereas the gradients for membrane proteins followed the segmentation borders (Fig. 2D). We, therefore, used guided back propagation to examine the gradients of the network at varying radii from the cell center. We found that for nuclear proteins, such as FoxP3 and SOX10, CellSighter turns its attention closer to the center of the cell whereas

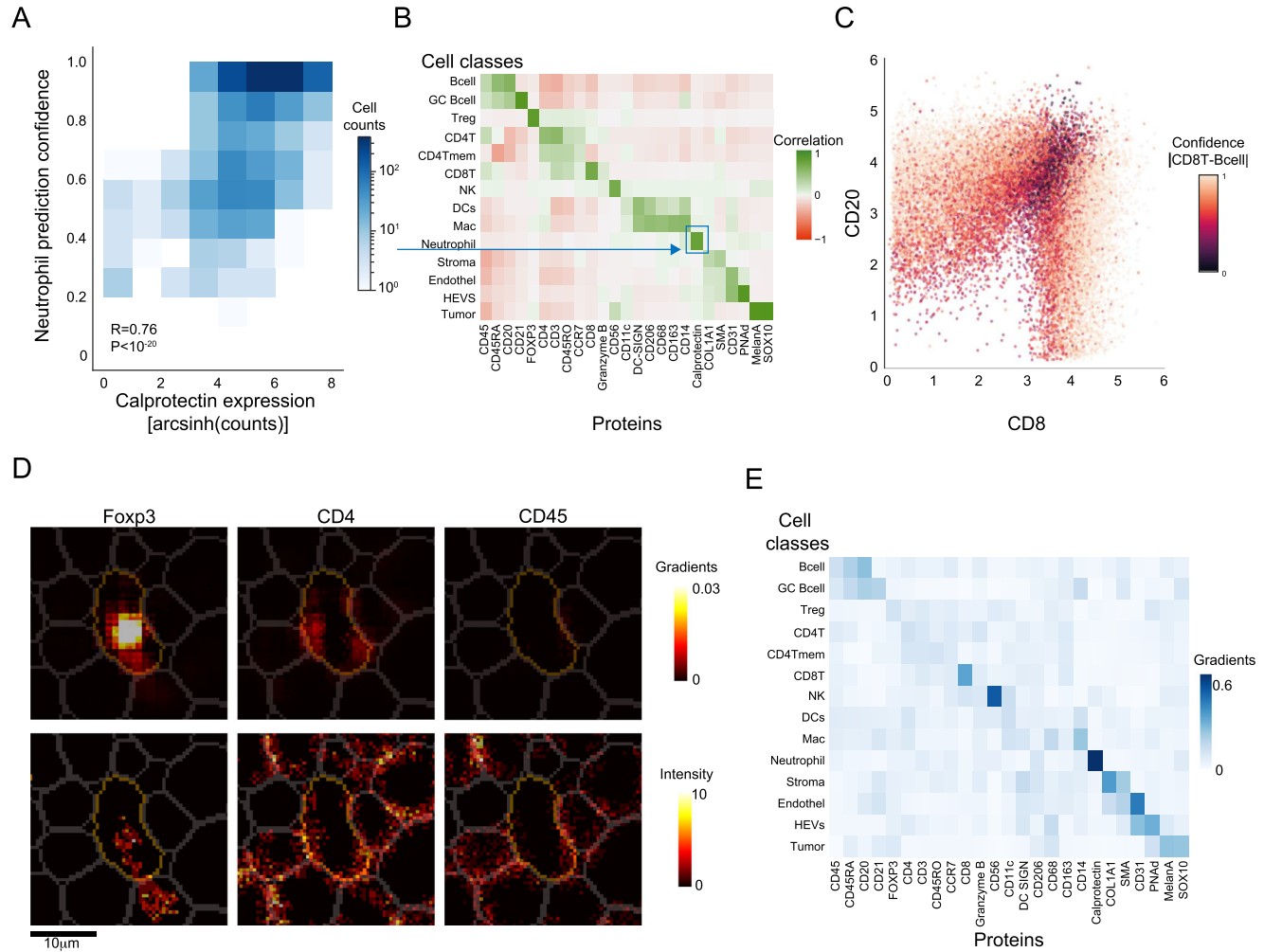

**Fig. 2 | CellSighter learns protein expression patterns. A** For all cells predicted as Neutrophils, shown is a 2D histogram depicting the correlation between the cellular expression levels of Calprotectin (x-axis) and CellSighter's confidence in predicting Neutrophil (y-axis). **B** For each protein (x-axis) in each cell type (y-axis), shown is the correlation between the cellular expression levels of this protein and the confidence in prediction of this cell type. High positive correlations are observed between lineage proteins and their cognate cell types. **C** Scatter plot of cellular expression levels of CD8 (x-axis) vs. CD20 (y-axis) for cells classified as either

cytotoxic T cells or B cells. Each cell is colored according to the absolute delta between its confidence for CD8T and Bcell. **D** Shown are the expression levels (bottom) and normalized gradients of the CNN (top) for a single cell classified by CellSighter as a Treg, in the images of Foxp3 (left), CD4 (middle) and CD45 (right). The images of the gradients show which proteins influence classification and where the network is looking for them. **E** Shown is the average positive gradients for each protein (x-axis) in each class (y-axis) normalized across the proteins, calculated from a single CNN on 4961 cells. Source data are provided as a Source Data file.

for membrane proteins this distance increases. For example, for FoxP3, 60% of the gradient is achieved at a distance of 40% from the center of the cell, whereas for CD4 it is at 75% (Fig. 3F and Supplementary Fig. 2G).

To examine whether this relationship was causal, we performed additional simulations. Again, we took patches centered on T helper cells and removed their cognate CD4 and CD20 signals, to avoid any confounding factors incurred by the original signals, and then reintroduced CD4 as a membranous signal. However, this time we also introduced CD20 as a membranous signal on the same cell. In different simulations we varied the percent of membrane that was covered by CD20, ranging from 100% to 12.5% (Fig. 3G). We also varied the overall signal in a complimentary manner, such that the average CD20 per cell was maintained at a relatively constant level (Supplementary Fig. 2H). Here, too we made sure that the overall signal was drawn from the real CD20 expression distribution (Methods). We found that when the CD20 signal surrounded 100% of the membrane, both models were equally likely to classify the cell as a B cell, resulting in 84% of the cells classified as B cells using CellSighter and 80% using XGBoost. However, as the fraction of overlap with the membrane was reduced, XGBoost

continued to classify a similar percentage of cells as B cells, whereas CellSighter was less likely to classify the cells as B cells. For example, at 12.5% overlap, XGBoost classified 72% of the cells as B cells, whereas CellSighter dropped to 35% (Fig. 3H). This suggests that CellSighter learned the characteristic membranous expression pattern of CD20.

Overall, we found that CellSighter learns both protein expression levels and spatial expression patterns and integrates both when classifying cells. We note that CellSighter was able to learn these spatial features even though it was trained on imperfectly-labeled data, where annotations were mostly generated using gating and clustering on the expression matrix. This is important because most labs who perform multiplexed imaging can relatively easily generate such imperfect annotations for a subset of the data, whereas generating high-quality manually-curated annotations is difficult and time-consuming. The fact that CellSighter learns spatial and sub-cellular expression features suggests that it is able to generalize beyond just learning the clustering.

**CellSighter features contribute to performance**

Next, we evaluated how different features of CellSighter affect the performance of the predictions. First, we examined what benefits, if

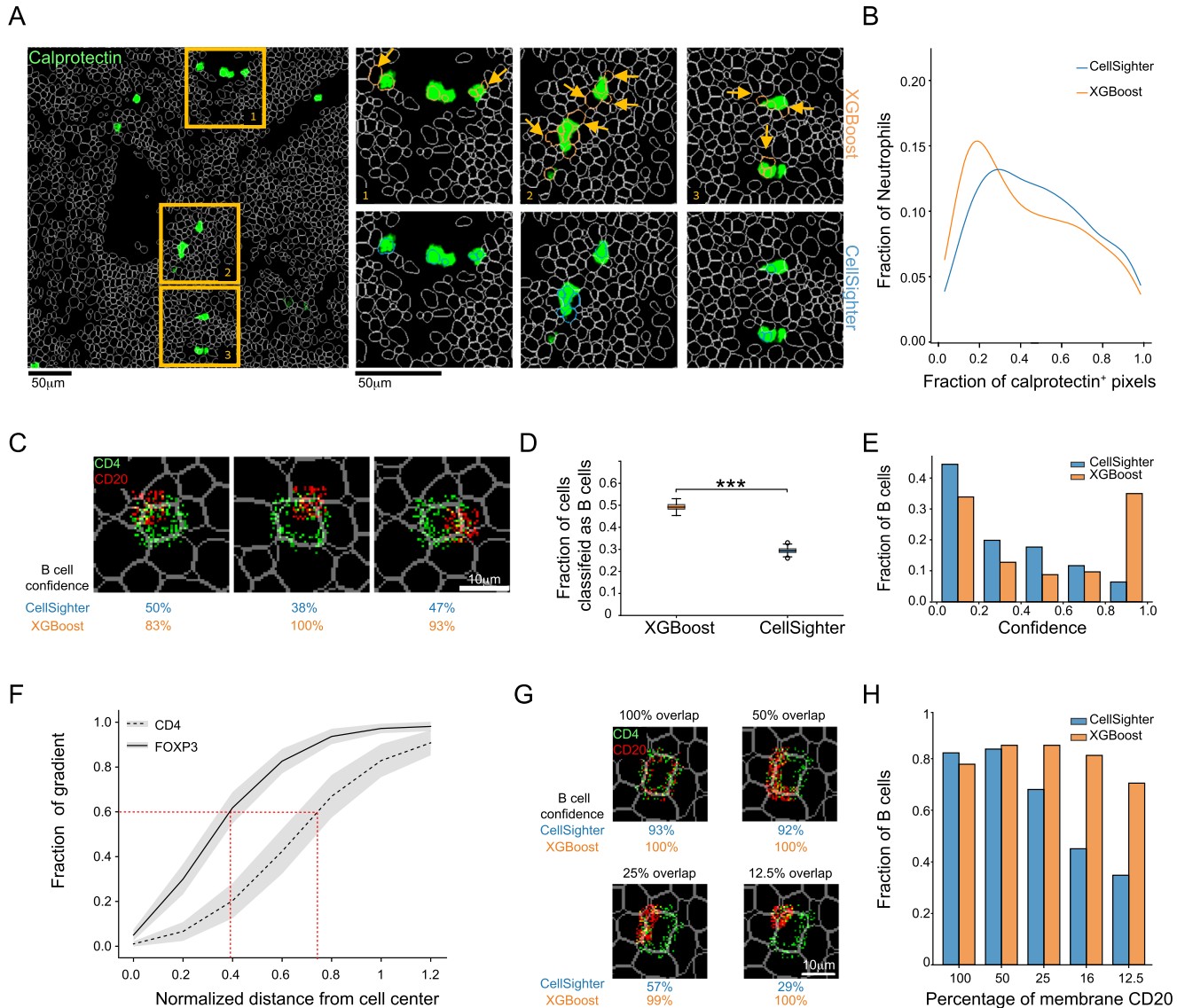

**Fig. 3 | CellSighter learns spatial expression features. A** Left: Calprotectin (green) and cell boundaries (white) in one FOV. Right: Zoom-in on boxes 1–3. Shown are cells classified as Neutrophils by XGBoost (top, orange) or CellSighter (bottom, blue). Arrows show cells that were classified as neutrophils by XGBoost, but not CellSighter. **B** The proportion of cells classified as neutrophils (y-axis) as a function of the fraction of pixels in the cell which stain for Calprotectin (x-axis). CellSighter (blue) classifies as neutrophils less cells with low overlap with Calprotectin. **C** Example images in which CD4 (green) is simulated as a membranous signal and CD20 (red) as a patch, partially overlapping with the cell membrane. **D** Simulations of 549 cells as in (**C**) repeated 100 times. Shown is the fraction of cells classified as B cells (y-axis) by XGBoost and CellSighter. Asterix denote a two tailed *t*-test, $P < 10^{-20}$. Boxplots show median, first and the third quartile. Whiskers reach up to $1.5 \cdot (Q_3 - Q_1)$ from the end of the box. Dots denote outliers. **E** Simulations of 549 cells as in (**C**). Shown is the fraction of cells classified as B (y-axis) as a function of the models'

confidence (x-axis). CellSighter (blue) has lower confidence in erroneous classifications relative to XGBoost (orange). **F** Shown is the normalized sum of gradients (y-axis) as a function of the normalized distance from the cell center (x-axis) for 1353 T-helper and 42 Tregulatory cells. Gradients for FoxP3 (nuclear protein, solid line) and CD4 (membranous, dashed line) reach 60% of their maximum at ±40% and ±75% of the cell, respectively. Mean ± (SD/2) are shown by lines and gray area. **G** Example images in which CD4 (green) is simulated as a membranous signal and CD20 (red) as a membranous signal with differential overlap of the membrane, ranging from 100% to 12.5%. **H** Simulations of 200 cells as in (**G**). Shown is the fraction of cells classified as B (y-axis) as a function of the percent of membranous CD20 (x-axis). Despite similar signal levels across the simulations, CellSighter (blue) was less likely to classify cells as B cells when the overlap with the membrane was small. Source data are provided as a Source Data file.

any, are incurred by using an ensemble of models. To this end, we correlated between the cellular protein expression levels and the confidence in prediction using either a single CNN or an ensemble. We found improved correlations using the ensemble. For example, using a single CNN 2.68% of the cells that were classified as Neutrophils had low expression of Calprotectin (<2), yet they were classified as Neutrophils with high confidence (>50%). Using the ensemble, this number drops 5-fold to 0.47%, and cells that are classified as Neutrophils either have high expression of Calprotectin or are classified with low confidence (Fig. 4A). Moreover, we compared the confidence in prediction

for all the cells for which our prediction agreed with the expert labeling to the confidence in prediction for all the cells for which our prediction disagreed with the expert labeling (Fig. 4B). We found that using a single model 20% of the cells that were wrongly classified had high confidence (>0.9). However, using an ensemble, this number dropped to 5%, and the overall confidence for wrong classifications was significantly lower (Fig. 4B). Overall, limiting the analysis to high-confidence cells, using a cutoff of 0.7 on the probability, results in classifications for 79% of the dataset and increases the recall of prediction by ~10% (Supplementary Fig. 3A). Neural networks can be

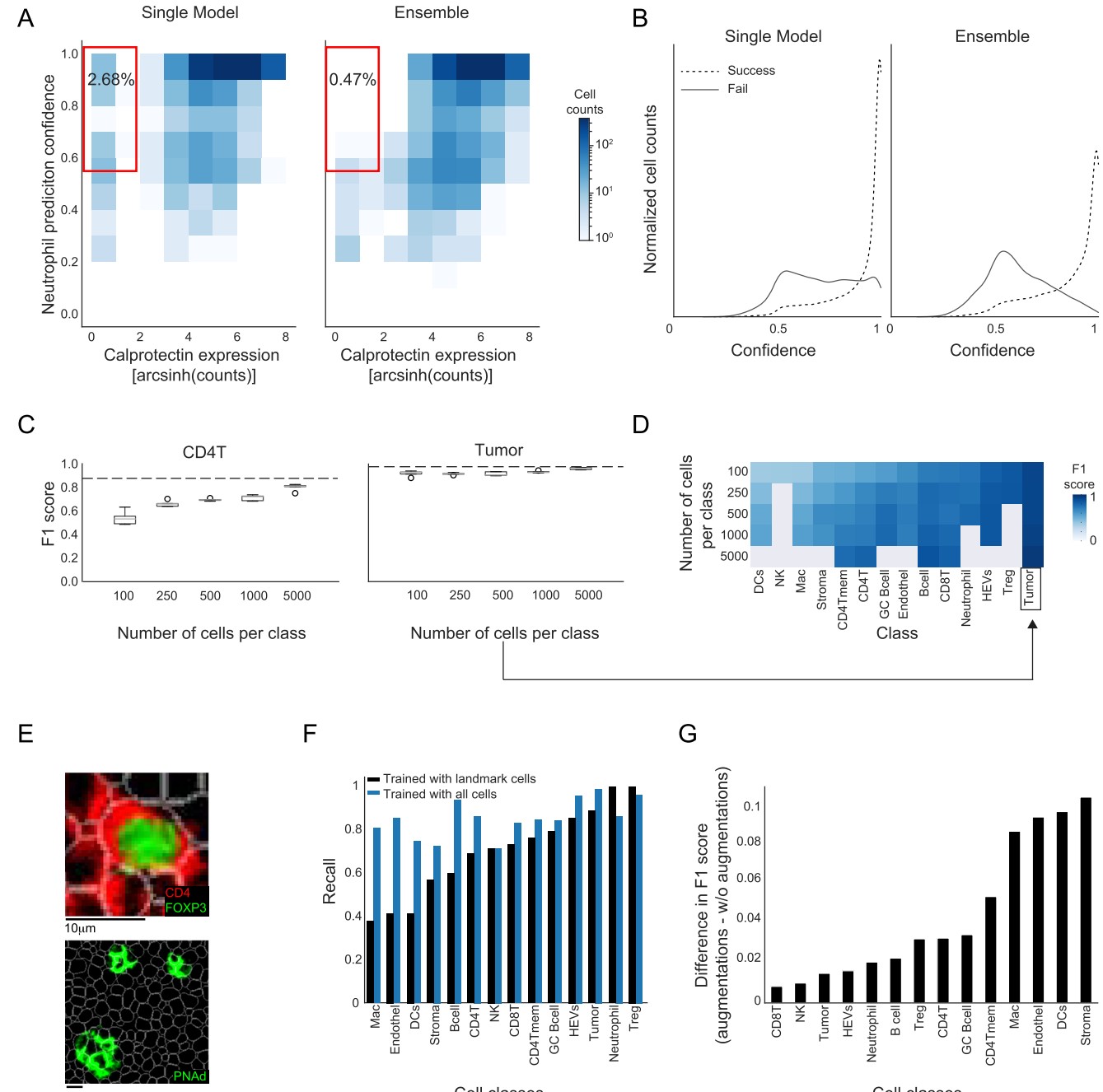

**Fig. 4 | Features of CellSighter contributing to performance. A** For all cells predicted as Neutrophils, shown is a 2D-histogram of the correlation between the expression levels of Calprotectin (x-axis) and CellSighter's confidence in prediction (y-axis). Using an ensemble of models (right) reduces the fraction of cells with low Calprotectin expression that are confidently classified as Neutrophils (red square) compared to a single model (left). **B** Shown is the fraction of cells (y-axis) for varying confidence levels (x-axis), for cells in which CellSighter predictions agree or disagree with expert labeling (solid and dashed lines, respectively). The ensemble (right) reduces the fraction of cells where CellSighter predicts with high confidence a class that differs from expert annotations, relative to a single model (left). **C** For CD4 T cells (left) and Tumor cells (right), shown is the classification accuracy (F1 score, y-axis) as a function of the number of cells used for training (x-axis). Boxplots show the results from five independent experiments with randomly sampled cells. Dashed lines show the F1 score on the entire training set. Boxplots show median, first and the third quartile. Whiskers reach up to $1.5 \cdot (Q_3 - Q_1)$ from the end of the box. Dots denote outliers. **D** Prediction accuracy (blue) for each class (x-axis) when varying the number of cells from that class in the training (y-axis). Gray indicates experiments that could not be performed since the dataset did not contain enough cells. **E** Examples of cell classes that are easily classified with a small training set, either containing nuclear signals (Treg, top) or organized into defined structures (HEVS, bottom). **F** Classification recalls (y-axis) for different classes (x-axis) using a model that was trained on landmark cells (black) or comprehensive annotations (blue). **G** Shown are the differences in accuracy of classification (F1 score, y-axis) for different classes (x-axis) between a single model trained with augmentations and without augmentations. The mean F1 score of epochs 35,40 and 45 is shown for each class. Source data are provided as a Source Data file.

difficult to train and can learn spurious correlations. Using an ensemble of models provides results that are more robust, with improved correlation between the network's confidence in prediction and its accuracy.

In addition, we evaluated how much data is needed to train CellSighter. To this end, we retrained CellSighter on training sets that varied in size, where we randomly sampled from each class either 100, 250, 500, 1000 or 5000 cells and evaluated the resulting accuracy in

prediction. To obtain confidence intervals, we repeated this process 5 times, each time training a separate model (Fig. 4C, D). We found that the number of cells needed to plateau the prediction accuracy was highly variable between classes. For example, for CD4 T cells we observed a continuous improvement in F1 score from 50% to 80% when increasing the number of cells. In contrast, for tumor cells, we found that increasing the training set resulted in only a modest increase in F1 score from 91% to 96% (Fig. 4C). These results indicate that some cell types are easier to learn than others. This can result from these classes having better defining markers, such as nuclear proteins that are less prone to spillover. Another factor contributing to making a class more easily classifiable could be spatial organization patterns where cell types of the same class cluster together, such as in the case of HEVs (Fig. 4E). Overall, these results suggest that labeling efforts can be prioritized in an iterative process. A user can label a few images and train CellSighter using either all cells or a subset to identify classes that would benefit most from additional training data. The CellSighter repository supplies functions to facilitate such analyses.

Encouraged by our results that showed that CellSighter could be trained on only thousands or even hundreds of cells, we checked whether it was possible to train the network on easy, well-defined cells, that don't suffer from segmentation errors, noise and spillover. If possible, this workflow would be highly advantageous as it would drastically reduce the time invested in the initial labeling. To this end, we identified in the dataset landmark cells that can be quickly defined using conservative gating (Methods). We then retrained CellSighter only on landmark cells from the 12 images in the training set. As expected, evaluating this model only on landmark cells in the test set achieves excellent results (recall $97 \pm 4\%$, Supplementary Fig. 3B). Next, we tested how this model performs on the entire test set. We found that using only landmark cells for training results in good classifications for well-defined classes such as Tregs, Tumor cells and HEVs. However, overall, it achieved poorer classifications relative to using an unbiased representation of the dataset (recall $70 \pm 20\%$ vs. $85 \pm 8\%$, Fig. 4F and Supplementary Fig. 3C). We conclude that simple gating could be sufficient for some cell types, but for others the network needs to be trained on representative data that reflects the issues in the real data. This information can be useful to prioritize labeling efforts to more difficult cells.

Finally, we evaluated the contributions of augmentations by training CellSighter with and without augmentations. We found that removing augmentations reduces the prediction accuracy by 1% to 10%, depending on cell type (Fig. 4G). Expectedly, the effect of augmentations was more significant for cell types that were under-represented in the training set and are difficult to classify, such as stromal cells and dendritic cells. Overall, having a wide plethora of augmentations diversifies the training set and helps overcome imbalances in the prevalence of different cell types.

We also tested the effects of other modifications to the pipeline, including different forms of data normalization (Supplementary Fig. 3D, E), inclusion of functional proteins in the classification process (Supplementary Fig. 3F), reducing the image resolution from (0.5 μm/pixel) to (1μm/pixel) (Supplementary Fig. 3G) and overclustering (Supplementary Fig. 3H–J). None of these materially affected the classification performance for this dataset. Reducing the resolution did reduce the time to train the model by 166% (from 5 min per epoch to 3 min per epoch), suggesting that resolutions of 1 μm/pixel may suffice for cell-level classification tasks.

### CellSighter generalizes across datasets and platforms

We evaluated whether CellSighter could apply to different datasets and platforms. First, we analyzed a published dataset of colorectal cancer acquired using CO-Detection by indEXing (CODEX)[26], a cyclic fluorescence-based method. We downloaded a publicly available dataset from Schurch et al., used Mesmer[20] to resegment the cells, and generated highly-curated ground-truth labels for 85,179 cells, from 35 FOVs using gating and sequential rounds of visual inspection and manual annotation. We trained CellSighter on 27 images encompassing 66,691 cells and tested the results on the cells from 8 heldout images. CellSighter achieved good levels of recall ($80 \pm 9\%$), indicating that the approach can also be applied to fluorescent data (Fig. 5A, B). Next, we evaluated a published dataset of Melanoma metastases acquired by Imaging Mass Cytometry (IMC), a different mass-based imaging modality[42]. We used the cell classifications provided by the authors to train CellSighter on 55 images and tested the results on 16 heldout images. CellSighter achieved high levels of recall ($84 \pm 17\%$) on all classes except for stroma (36%), which was mostly confused with tumor cells (Fig. 5C).

Finally, we checked whether a model trained on one dataset could be applied to a different dataset. This is a challenging task since different datasets are collected on different tissues where different populations of cells reside, and these cells may have altered morphology, phenotypes and spatial organizations. Technically, different datasets will typically differ in the number and identity of proteins visualized and may have batch effects relating to the instrumentation and antibodies used at different times. We evaluated how CellSighter, trained on the melanoma lymph node dataset performed on a dataset profiling the gastrointestinal tract, which shares 19 proteins used to define eight shared cell types (Fig. 5D). We found that training Cell-Sighter on the melanoma lymph node data and evaluating on the gastrointestinal data achieves high results for major cell types that shared all of their defining proteins, including CD4 T cells (88%), Tregs (83%), CD8 T cells (93%) and Endothelial cells (82%). For macrophages the performance was significantly low (47%). Notably, for this class there were unique proteins which were used for expert labeling in one dataset, but not the other. As such, poorer performance could stem from not having enough information for classification, or indicate biological variability in the expression patterns and morphology of these cell types between the lymph node and the gut[43].

Altogether, we conclude that CellSighter achieves accurate cell type classification within a single dataset for different types of multiplexed imaging modalities. Across datasets, accurate classification is dependent on having shared lineage-defining proteins, morphology and phenotypes.

## Discussion

Cell classification lies at the heart of analysis of multiplexed imaging, but has hitherto remained labor-intensive and subjective. In this work, we described CellSighter, a CNN to perform cell classification directly on multiplexed images. We demonstrated that the network learns both features of protein co-expression as well as spatial expression characteristics, and utilizes both to drive classification. We showed that CellSighter achieves high accuracy (>80%), on par with current labeling approaches and inter-observer concordance, while drastically reducing hands-on expert labeling time.

CellSighter has several features that we found appealing as users who frequently perform cell classification on multiplexed images. First, CellSighter outputs for each cell not only its classification, but also a confidence score. This type of feedback is nonexistent using current clustering or gating approaches, which often result in variable and arbitrary quality of cell classifications. Working with CellSighter, we found that the confidence scores that are generated are useful in evaluating any downstream analyses that are based on these classifications. Furthermore, evaluating CellSighter's predictions on a test set is highly informative of label qualities. We consistently found that improving the labeling that is used for training improves CellSighter's performance and ability to generalize. Therefore, classes that have low prediction accuracies can help the user to identify cell types that are poorly defined. This, in turn, can direct further efforts to split classes, merge classes, gate, or perform manual annotations on the cells of the

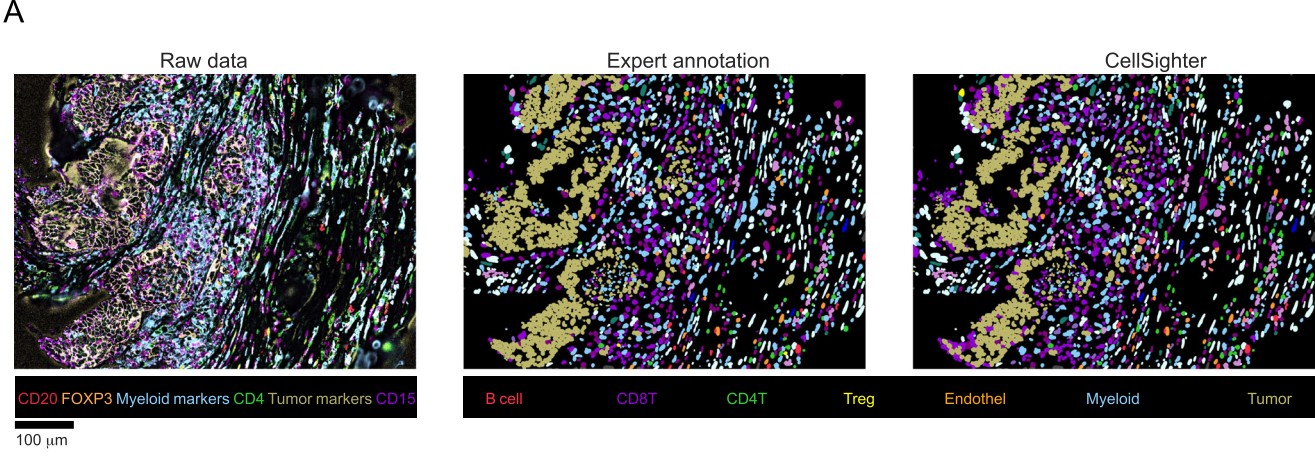

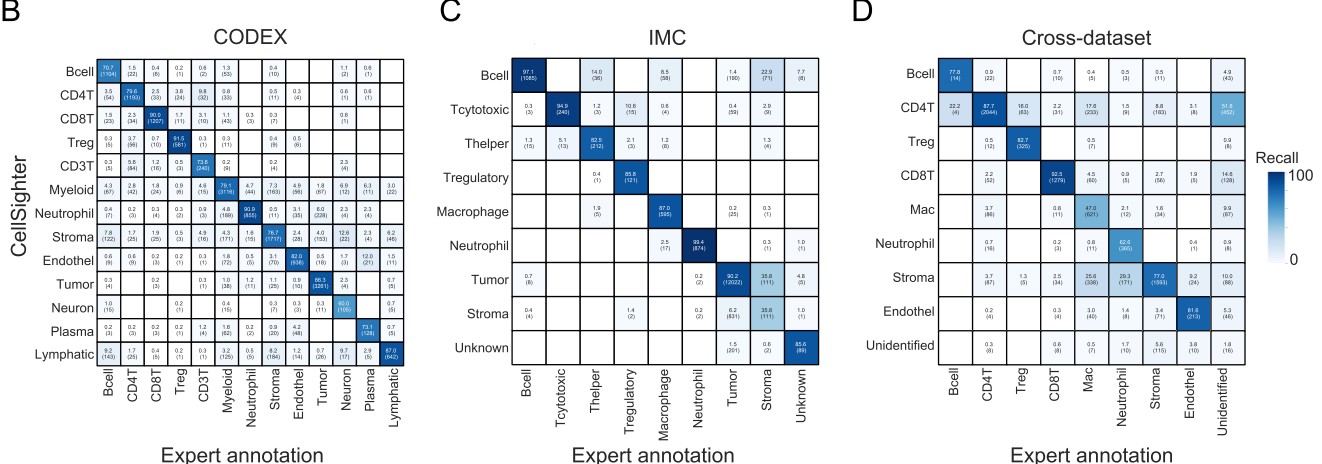

**Fig. 5 | CellSighter generalizes across datasets and platforms. A** For one FOV in the CODEX dataset from Schurch et al., the protein expression levels (left), expert-generated labels (middle) and CellSighter labels (right) are shown (**B**) Comparison between labels generated by experts (x-axis) and labels generated by CellSighter (y-axis) for the CODEX dataset from Schurch et al. (**C**) Same as (**B**) for IMC data from ref. [42] **D** Same as (**B**) showing performance of a model trained on a MIBI melanoma lymph node metastases dataset and evaluated on a dataset of the gastrointestinal tract. Source data are provided as a Source Data file.

training set until adequate results are achieved. This process will likely increase the overall labeling quality in multiplexed imaging.

CellSighter also has some limitations. Primarily, it is a supervised approach. As such, if a rare cell population (eg Tregs) is not represented in the training set, CellSighter will not be able to identify it in the rest of the dataset. One way to diminish this issue is to validate that for each antibody there are images containing positive staining in the training set. Still, this will not resolve rare populations that are defined based on differential combinations of proteins that are usually associated with more abundant populations. For example, rare FoxP3[+] CD8[+] T cells[44] may not be included in the training set and therefore missed. Incidentally, such extremely rare populations are also commonly not classified using standard clustering and gating-based approaches[26,27,42]. With CellSighter, such cells can be more readily identified by examining cells that were confused between classes. For example, FoxP3[+] CD8[+] T cells could be identified by evaluating cells that have high probabilities to be classified as either Tregs or CD8 T cells. The user can then decide whether to add this class and make sure that it is represented in the training set or, alternatively perform subsequent gating of this population.

In addition, CellSighter receives as input images of multiple channels. This has a big advantage in that multiple proteins are assessed simultaneously to call out cell types. For example, CD4 T cell classification will be driven by expression of CD3 and CD4, but not CD8 and assessing these proteins for both the classified cell and its immediate surroundings. This mimics what human experts do when they perform manual labeling and adds to the accuracy of classification. On the flip side, different datasets often include different proteins in their panels. A good example for this is myeloid cells, where different studies measure different combinations of CD14, CD16, MHCII, CD163, CD68, CD206, CD11C, etc[6,26,27,42,45–48]. Transferring models between datasets that don't share all the proteins used in classification is not straightforward, and reduces the accuracy of classification. There are several solutions to this issue. We found that proteins can be interchanged if they share a similar staining pattern. In addition, as technologies mature, antibody panels will likely increase in size and become more standardized, reducing inter-dataset variability in the proteins measured, which will facilitate transferring labels across datasets and platforms. Altogether, we foresee that in the future, using machine-learning approaches such as CellSighter will streamline data integration, such that knowledge would transcend any single experiment and consolidate observations from different studies[49,50]. Augmenting cell-based classification with pixel-level classifiers[33,51], also stands to provide benefit and increase accuracy. While CellSighter is undoubtedly not there yet, it is an important step in facilitating this process.

## Methods

### Datasets and expert annotations

**MIBI melanoma dataset.** 111 $0.5 \times 0.5$ mm$^2$ images, altogether encompassing 145,668 cells. Labels for all cells were generated using sequential gating and sequential rounds of visual inspection and extensive manual annotation. Of these, the cells from 101 images were used for training and the model was evaluated on the remaining 10 images. The following populations have been defined: B cells (CD20, CD45), CD4 T cells (CD3, CD4, CD45), CD8 T cells (CD3, CD8, Granzyme B, CD45), Endothelial (CD31), Myeloid (CD16, CD68, CD163, CD206, DC_SIGN, CD11c, CD45), Mesenchyme (SMA), Neutrophil (MPO_ Calprotectin), T cell (CD3 only), T regs (FoxP3) and Tumor (SOX10). The other cell types in this dataset were not used as they have ambiguity in their definition. Expression of all proteins across cell types can be found in Supplementary Fig. 4A.

**MIBI melanoma lymph node dataset.** The dataset contains sixteen labeled $0.8 \times 0.8$ mm$^2$ images, altogether encompassing 116,808 cells; and 33 unlabeled images, altogether encompassing 333,765 cells. Of these, the cells from 12 labeled images were used for training and the model was evaluated on the remaining four labeled images. Labels for all cells were generated using FlowSOM clustering[24], in combination with gating and sequential rounds of visual inspection and manual annotation. The following populations have been defined: B cells (CD20, CD45, CD45RA), DCs (DC_SIGN, CD11c, CD14, CD45, CCR7, CD4), CD4 T cells (CD4, CD3, CD45), T regs (CD4, FoxP3, CD3, CD45), Macrophages (CD45, CD68, CD163, CD206, DC_SIGN, CD14, CCR7), CD8 T cells (CD8, CD3, CD45, Granzyme B), Stroma (COL1A1, SMA), Follicular Germinal B cells (CD20, CD21, CD45, CD45RA,), HEVs (CD31, PNAd), Memory CD4 Tcells (CD4, CD3, CD45, CD45RO), NK cells (CD45, CD56), Neutrophils (S100A9_Calprotectin), endothelial cells (CD31) and Tumor (MelanA, SOX10). Additional 8 cells were labeled as immune cells. For some of the experiments on this dataset the following expert-annotated classes were not included in predictions: unidentified, which contains a mixture of various proteins, and CD3-only, together encompassing 5% of the data.

Expression of all proteins across cell types can be found in Supplementary Fig. 4B.

**MIBI gastroIntestinal (GI) dataset.** Labels for eighteen $0.4 \times 0.4$ mm$^2$ images were generated using FlowSOM clustering[24], in combination with pixel clustering[33] and sequential rounds of visual inspection and manual annotation. The model trained on the melanoma lymph node dataset was ran on 9,532 cells from the following classes, which were shared across the two datasets: T regs (FoxP3, CD4, CD3, CD45), CD8 T cells (CD8, CD3, GranzymeB, CD45), CD4 T cells (CD4, CD3, CD45), B cells (CD20, CD45RA, CD45), Macrophages (CD68, CD206, CD163, CD14, DC-SIGN, CD45), Neutrophil (S100A9-Calprotectin), Stroma (SMA, COL1A1) and Endothelial (CD31). Expression of all proteins across cell types can be found in Supplementary Fig. 4C.

**IMC melanoma dataset.** Data and cell classifications were taken from ref. [42]. Of these, the cells from 55 images were used for training and the model was evaluated on the remaining 16 images, altogether encompassing 70,439 labeled cells. The following populations, as defined by the authors, have been used: Tumor, B cell, CD4 T cells, Macrophage +pDC, CD8 T cells, Stroma, Neutrophil, Tregs and unknown. Annotations were provided for a subset of cells. For this dataset, the crop size for the CNN was chosen to be $30 \times 30$ pixels because of the image resolution. To train CellSighter the following subset of protein channels was used: CD4, CD20, SMA, SOX10, FOXP3, CD45RO, Collagen I, CD11c, CD45RA, CD3, CD8a, CD68, CD206/MMR, S100, CD15, MPO, HLA-DR, CD45, CD303, Sox9, MiTF, CD19, p75.

**CODEX colorectal dataset.** Data were taken from ref. [26]. We used Mesmer to resegment the cells, and generated highly-curated ground-truth labels for 85,179 cells, from 35 FOVs using gating and sequential rounds of visual inspection and manual annotation. Cells were classified to the following classes: Myeloid, CD4 T cells, CD8 T cells, T regulatory cells, CD3T cells (CD3$^+$ only), Neutrophil, Stroma, Endothelial, Neuron, Lymphatic, Plasma cells, B cells, and Tumor cells. Expression of all proteins across cell types can be found in Supplementary Fig. 4D. The cells from 27 images were used for training and the model was evaluated on the remaining 8 images. CellSighter was trained on the following proteins: CD11b, CD11c, CD15, CD163, CD20, CD3, CD31, CD34, CD38, CD4, CD45, CD56, CD57, CD68, CD8, Collagen, Cytokeratin, FOXP3, HLADR, MUC1, NAKATPASE, PDPN, SYP, VIM, and SMA.

### CellSighter

CellSighter is an ensemble of CNN models, each based on a ResNet50 backbone[52]. The input for each model is a 3-dimensional tensor, consisting of cropped images of K proteins centered on the cell to be classified, a binary mask for the cell to classify (1's inside the cell and 0's outside the cell), and a similar binary mask for all the other cells in the environment (Fig. 1C). The crop size can vary, but ideally should include the cell and its immediate neighbors. For the datasets at hand no significant differences were observed when varying the crop sizes from 40–100 pixels (corresponding to $\pm 20$–50 µm$^2$, see Supplementary Fig. 1C). A crop size of $60 \times 60$ pixels was used for all datasets except the IMC, where a crop size of $30 \times 30$ pixels was used to account for the different resolution.

To account for class imbalance, rare classes were upsampled, either equally or such that the major lineages (Myeloid, T cell, tumor, B cell, other) were represented in equal proportions. *Training*: In training, the images randomly undergo a subset of the following augmentations: no augmentation, rotations, flips, translations of the segmentation mask, resizing of the segmentation mask by up to 5 pixels, shifts of individual protein channels in the X-Y directions by up to 5 pixels with probability of 30%, and gaussian signal averaging in a window of 5 pixels followed by Poisson sampling. The final cell classification is given by integrating the results from ten separately-trained CNNs. We found that randomization in initializations and augmentations are sufficient to generate sufficient diversity between the models. Each CNN performs multi-class classification and outputs a probability vector for all classes. Those probabilities are then averaged to generate one probability per cell type. The final prediction is the class with the maximal probability and the prediction confidence is the probability value.

The code for CellSighter can be found at: https://github.com/KerenLab/CellSighter.

### CellSighter running times

Preprocessing of the dataset to crops is performed once per dataset, and takes a few minutes for 250,000 cells. Training one model in the ensemble with one GPU (Quadro RTX 6000) for one epoch, which is equal to one pass over whole train set (132,052 cells), took approximately 6 min for the Melanoma dataset. Each model in the ensemble can be trained in parallel. For the different datasets in the manuscript, we trained for between 10 to 40 epochs, depending on the dataset. So altogether, training on one GPU took 1–4 h. Evaluation on new data takes approximately 40 s for 1000 cells, so ~7 min for 100,000 cells. Inference can easily be done in parallel to reduce running times. Naturally, the number of cells for training, crop size, batch size, RAM availability and other parameters can affect the running time.

### XGBoost

Python's XGBoost gradient boosting tree model was used for benchmarking experiments on tabular data (https://github.com/dmlc/

xgboost[39]). Input to the model included the arcsinh transformed expression values per cell normalized by cell size for all the proteins that were used to train CellSighter[20]. For some of the datasets balancing of cell types was implemented to increase performance. The model was trained with the following parameters: n_estimators = 100 and max_depth = 2, the rest of the parameters were kept as the default of the library.

### Gradient analysis

We can refer to CellSighter as a complex function $F$. Given an Image $x$ we can compute $F(x)$ to be the cell classification, which is a vector with probability per cell type. We can then derive $F$ by $x$ and see how each pixel in $x$ affects the value of $F$. For each pixel and cell type, we get a value that indicates how important it is for predicting the cell to be of a specific cell type.

The analysis was performed using one of the models of the ensemble on 5000 cells from the test set. Small cells (<50 pixels) were removed since their size is too small for spatial analysis, leaving 4961 cells. For each of these cells, the sum of positive gradients for each marker was calculated in concentric circles centered in the middle of the input, ranging from 1 to 18 pixels in jumps of 2. For each cell, the radius of the circles is then normalized by the radius of the cell. Gradients are normalized relative to the largest radius profiled. Guided back propagation was preformed using publicly available code[41].

### Neutrophil analysis

**Calprotectin patch analysis.** Calprotectin patches from seven unlabeled FOVs, overall including 1334 cells that were classified as Neutrophils, were obtained by dilating the signal using a kernel of size 3 and identifying connected components. Cells that overlap with Calprotectin patches are identified.

**Simulations.** All 237 Neutrophil cells from the test set were used. Calprotectin signal was simulated using Poisson sampling with lambda = 4 normally distributed with a standard deviation of 5 pixels. Simulations were performed in which this signal was moved relative to the center of the cell ranging from 0 to 15 pixels in x and y directions.

### CD4/CD20 simulations

**CD20 patch experiments.** 549 CD4 T cells were randomly-sampled and their cognate CD4 and CD20 signals removed to avoid any confounding factors incurred by the original signals. CD4 signal was simulated by Poisson sampling with lambda = 1.3 in horizontal and vertical distances of at most 5 pixels from the border of the cell segmentation. CD20 was simulated around a random point on the border of the cell with Poisson sampling with lambda = 1.3 with a uniform distance between 0 and 6. Cells that are smaller than 15 pixels across their minor axis were filtered to eliminate complete overlap of the patch with the cell.

**CD20 membrane experiments.** 200 CD4 T cells were randomly-sampled and their cognate CD4 and CD20 signals removed to avoid any confounding factors incurred by the original signals. CD4 signal was simulated by Poisson sampling with lambda = 1.3 in horizontal and vertical distances of at most 5 pixels from the border of the cell segmentation. CD20 was simulated similarly, but varying the percent of the membrane that is covered by the signal to be 12.5%, 25%, 50% and 100%. In order to keep the overall signal intensity in the cell similar, the number of sampled points increased proportionally to the decrease in membrane size.

### Training with different input sizes

For these experiments, 100, 250, 500, 1000 or 5000 cells were randomly sampled from the training set for each class. In cases in which there were not enough cells in the data for sampling (e.g., Tregs had only 440 cells in the training set), all cells from that class were sampled, but results for these values for these classes are not reported to allow comparisons across experiments. Figures show the mean and std of five independent experiments.

### Landmark cell analysis

Landmark cells were defined for each class as the cells that express above 20th percentile of the proteins that define the class, and are not above the 15th percentile value of expression of any other protein. E.g., landmark tumor cells strongly expressed either SOX10 or MelanA and no other lineage protein. For the following classes some deviations from this formulation were necessary to allow enough cells for training: For myeloid cells the threshold for the other markers was the 20th and not 15th percentile. For Tregs and neutrophils only high expression of FoxP3 and Calprotectin was used, respectively, without consideration for other proteins. Visual inspection validated that these cells were indeed landmarks of their classes.

Landmark cells were partitioned to train and test by the same image partition as in Supplementary Fig. 2A. CellSighter was trained on the landmark training cells and tested both on the landmark test cells (see Supplementary Fig. 3B) and on all cells in the test (see Supplementary Fig. 3C).

### Normalization

Two normalization methods were tested for their effect on classification. The first method is Anscombe transformation, which reduces the effect of the heavy tails in the distribution, and is commonly used for low-count images. The second was scaling each pixel by the 99th percentile. Results for both normalizations are shown in Supplementary Fig. 3D, E.

### Over-clustering

To assess whether over-clustering improves CellSighter predictions, the model was trained on two sets of labels. The first set included lineage classes "DCs" and "Macrophages". In the second set, these classes were over-clustered to subsets, including DCs, CD14+ CD11c+ DCs, CD11c + DCsign+ DCs, Macs, Mono CD14+ DR, CD68+ Mac, DC-SIGN+ Mac, and CD206+ Mac. Supplementary Fig. 3H depicts the expression profiles of these subsets. CellSighter was trained on both sets of labels and the results for "DCs" and "Macrophages" were evaluated on the test set (see Supplementary Fig. 3H–J).

### Image resolution

To investigate the effect of image resolution on classification results, we took the Melanoma LN data, which was acquired at a resolution of 0.5 μm/pixel and then simulated from this data images at a resolution of 1 μm/pixel using a 2 × 2 kernel (see Supplementary Fig. 3G).

### Assessment of the contribution of functional proteins for cell classification

To assess the contribution of functional proteins for cell classification, CellSighter was trained on the same dataset using either 25 lineage proteins, as described above, or 39 lineage + functional proteins, including: CD103, Bax, HLA-DR-DP-DQ, HLA-class-1-A-B-C, IDO-1, Ki67, LAG-3, PD-1, TCF1TCF7, CD45RA, TIM-3, Tox-Tox2, PD-L1 and CD69 (see Supplementary Fig. 3F).

### Image visualization

For visualization purposes only, images were clipped to the dynamic range. Some MIBI images were Gaussian blurred using ImageJ (https://imagej.nih.gov/ij/download.html). Figures were prepared using Biorender and Adobe Illustrator.

## Statistics and reproducibility

All statistics were computed using Python 3.8.5. A complete list of packages can be found at: https://github.com/KerenLab/CellSighter/blob/main/requirements.txt. Mesmer's version 0.11.1 and later were used.

## Reporting summary

Further information on research design is available in the Nature Portfolio Reporting Summary linked to this article.

## Data availability

Melanoma lymph node dataset is available at: https://doi.org/10.17632/zcz8743fcv.2.

CODEX colorectal dataset is available at: https://doi.org/10.7937/tcia.2020.fqn0-0326.

IMC Melanoma data is available at: https://doi.org/10.5281/zenodo.6004986 Source data are provided with this paper.

## Code availability

The code for CellSighter can be found at: https://github.com/KerenLab/CellSighter.

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

## Acknowledgements

The authors thank Tal Keidar Haran, Eli Pikarsky, and Michal Lotem for samples and data of melanoma lymph metastases. L.K. holds the Fred and Andrea Fallek President's Development Chair. She is supported by the Enoch foundation research fund, the Abisch-Frenkel foundation, the Rising Tide foundation, the Sharon Levine Foundation and grants funded by the Schwartz/Reisman Collaborative Science Program, European Research Council (94811), the Israel Science Foundation (2481/20, 3830/21) within the Israel Precision Medicine Partnership program and the Israeli Council for Higher Education (CHE) via the Weizmann Data Science Research Center. I.M. is supported by a EU - Horizon 2020 - MSCA Individual Fellowship (890733). S.B. is a Robin Chemers Neustein AI Fellow and acknowledges funds from the Carolito Stiftung and the NVIDIA Applied Research Accelerator Program.

## Author contributions

Y.A. and B.F. developed CellSighter's architecture; Y.A. led the work, performed analyses and prepared figures; Y.B. and I.M. curated the data and prepared figures; S.B., I.M., and L.K. supervised the work; L.K. initiated the work; Y.A., Y.B., S.B., I.M., and L.K. conceived, designed, and interpreted analyses; Y.A., I.M., and L.K. wrote the manuscript.

## Competing interests

The authors declare no competing interests.
