## [Peer Review File · Nature Communications]

REVIEWER COMMENTS

Reviewer #1 (Remarks to the Author):

In this manuscript, Amitay et al describe a method for classifying cell types directly from multiplexed image stacks of in situ markers for different proteins. This analysis replaces a two-step process (extracting an expression vector for each cell, then classifying each cell based on this vector) with a single process, which has the potential to use local image context to determine a cell's type. This method is significantly more involved, more black boxy and more computationally expensive than the standard two-step process, so it should be definitively demonstrated that it is worth the trouble by showing that it performs better than the two-step process. Unfortunately, this does not seem to be the case, and the paper appears to hide this fact in a supplementary figure.

Most papers have two types of figures/analyses: the main analyses, which show that something works significantly better than prior art, and some secondary analyses, which show why it works better, how it can be applied, how performance varies with parameters etc. This paper does not have the main analyses, but it has secondary analyses and it does those really well. A lot of detail is put into carefully determining if the CNN indeed takes advantage of context to do its classification for example, which I think is convincingly proved. However, that analysis is only valuable if it can first be shown that the new proposed method works significantly better than the two-step process, at least in some cases. The only mention of this is somewhere in the middle of the paper, where it is said that "XGBoost (...) got high prediction accuracies, globally comparable to CellSighter (Supp Fig. 2A)". This should be the main figure, and unfortunately the only thing it proves is that CellSighter is not worth the trouble over the much simpler, much faster, and more interpretable two-step process.

I would suggest that the authors dig a lot deeper into making this comparison and try to understand why they couldn't beat the two-step XGBoost method. Their new approach is theoretically very interesting and promising, but something appears to be holding them back from being able to beat XGBoost. It is shown in some of the secondary analyses that XGBoost does not do as well as CellSighter at incorporating context information (because it does not see any context), which would imply that CellSighter loses performances elsewhere. Where? Perhaps if that can be investigated and fixed, CellSighter would be able to consistently outperform XGBoost.

Assuming the authors can convincingly beat XGBoost, I would be happy to review a revised version of this paper. The revised paper should include comparisons with the two-step method (XGBoost) for each quantitative figure, and the result across cell classes should be summarized with single scores and compared between methods. I really do hope the authors can beat XGBoost, because the method is theoretically very interesting, despite being complex. There should also be a comparison of processing speed. CellSighter requires a neural network to be run for each and every cell in the image, unless I misunderstood. Since many datasets are in the many thousands or millions of cells, this can quickly lead to prohibitively large runtime costs.

Reviewer #2 (Remarks to the Author):

Multiplexed tissue profiling constitutes one of the cutting-edge approaches to studying the tumor microenvironment as it provides spatial context to single-cell epitope co-localization and therefore cells' functional status. Although this is a promising technology, several technical problems and time-consuming barriers limit its application in clinical practice.

In this work, authors implement a deep-learning workflow, CellSighter, to streamline cell classification from multiplex imaging to reduce labor-intensive work and to improve on current expression matrix-based cell classification. This neural network approach provides an effective and efficient cell classification with a limited training set, which seems to be less susceptible to some of the bias related to current methods, including spillover. In my view, this method is of great interest to the field as it could allow a wider application of multiplex technology.

The article and abstract are overall clear, well written, and detail the various experiments conducted by the authors. References are well-detailed. I did not identify any major flaws in the

methodology and statistical results, which are in my view original, well-conducted, and reported. The conclusions provided are in line with the findings, robust, valid, and reliable, and do not exaggerate the benefits provided by this approach.

I'd like the author to specify two aspects:

- The authors tested this approach also on fluorescence data reporting a lower performance on "several classes", without detailing the problems. IF-based approaches are more diffuse than MIBI but are affected by several imaging problems which could greatly benefit from a neural-network assisted interpretation. I'd like, if possible, to more clearly detail the reason of the low performance and the limits in evaluating a fluorescence imaging with this CNN methods.
- They mention that the CNN model was trained on 10 separate-trained CNN, which should refer in my understanding to 10 cell lines. In methods I find a referral to 14 cells lines (B cells, DC, CD4, Treg, Macrophages, Cd8, Stroma, Germinal B cells, HEV, CD4 memory, NK, Neutrophils, Endothelial, Tumor). If that's the case, I'd like this aspect to be more clearly stated.

Reviewer #3 (Remarks to the Author):

The manuscript by Amitay et al. describes a novel approach, termed CellSighter, that employs neural networks to classify cell types in multiplexed imaging data. The authors demonstrate that CellSighter reproduces cell type labels obtained with currently employed methods, that these classifications are based on protein co-expression as well as the subcellular location of the proteins and they show how manipulating features of this framework impacts predictions. Lastly, they show that CellSighter can be applied to datasets from various technologies.

I believe that the manuscript constitutes an important advance in the field of spatial biology and multiplexed imaging and their approach to incorporate the imaging data for the task of cell classification is novel. Most of the claims in the paper are supported well, with the exception of my comments below. The work seems to be of high quality overall and appropriate methodologies were used.

Comments:

The manuscript mentions in several places that CellSighter would accelerate or speed up the process of cell classification. I am not sure this is true, and I don't think this is shown. In my understanding, to use CellSighter one still has to provide training data for which one either has to manually identify cells or alternatively go through the standard pipeline of segmentation and clustering. So overall, these time-consuming steps still seem part of the pipeline. The authors suggest that "It is easier and faster to classify thousands of cells than to classify millions of cells." but I am not fully convinced. Of course, clustering algorithm runtimes go up a bit, but many widely used algorithms can process millions of cells within minutes. Even in "manual gating" one often looks at all cells at once and cell numbers do not drastically influence the time it takes to identify cell populations. I suggest the authors remove this claim from the manuscript or show a convincing analysis of how their proposed workflow saves time.

The analysis shown in Figure 1 indicates that CellSighter accurately reproduces cell labels that were generated in more established ways, including through iterative clustering. As expected, there is some disagreement between the two. However, I am surprised that for these cells with disagreeing labels, it was not clear that CellSighter had produced more accurate labels. Does this imply that FlowSOM clustering, which I believed was used here, works equally well as CellSighter? I find this to be at odds with the later analysis showing superior performance of CellSighter compared to a machine learning classification approach. How do the authors explain this?

On the technical side, in many clustering pipelines it is common that integrated single-cell protein expression levels are normalized (e.g. asinh or log) and scaled. Was this done before the imaging data was fed into CellSighter? Furthermore, what would be the effect of channel-to-channel normalization, e.g. using the 99th percentile? Could this be used to improve performance?

I think it would be important to show how the number of clusters (which still has to be chosen manually) affects lineage identification accuracy. Is there any value in "over-clustering" the data

followed by pooling of some clusters to arrive at biologically meaningful populations? Or would this reduce overall accuracy?

It is great to see that CellSighter can be applied to different multiplexed imaging technologies. I am wondering whether the authors explored why the performance was lower on the CODEX dataset compared to MS-based technologies. Would this be related to fluorescence as a readout or is this specific to this particular dataset? This would be important information for the community.

Related, it would be much appreciated to show how imaging resolution affects cell classification, given the differences in resolution provided by different approaches to multiplexed imaging.

Given the issues with technical and biological factors as laid out in the manuscript: Is there a way to express prediction confidence on the pixel level? While not critical, I think this would be a huge improvement in the field to give users a way to disregard such overlapping or otherwise confounding pixels from the subsequent analysis.

It seems that the presented analysis was performed using commonly employed "lineage-markers" that mostly are expressed on one or few cell populations. Given that these technologies are able to quantify about 40-50 proteins, is there any value in including more proteins for the CellSighter analysis or can this also negatively impact classification performance?

Minor comments:

Not being an expert on CNNs, I was not familiar with the specifics of the term gradient. I think the concept should be introduced shortly in the manuscript, given the broad readership of Nature Communications.

Fig 2B what is the color scheme showing?

RESPONSE TO REVIEWERS' COMMENTS

We thank the reviewers for their in-depth consideration of our manuscript and for their insightful comments and suggestions. We were pleased to find all reviewers were very positive about the importance of the technological challenge that we were addressing, the novelty of the approach, the clarity of the results and writing and the potential impact of the work. We thoroughly addressed the reviewers' comments below by adding additional data, analyses, supplementary figures, tables, and references. Below we provide detailed point by point comments to the reviewers' remarks. We also supply a revised manuscript in which changes are marked by red fonts.

Altogether, we have comprehensively revised the manuscript and believe that it is materially improved. We thank the reviewers for their time and contribution.

Reviewer #1 (Remarks to the Author):

In this manuscript, Amitay et al describe a method for classifying cell types directly from multiplexed image stacks of in situ markers for different proteins. This analysis replaces a two-step process (extracting an expression vector for each cell, then classifying each cell based on this vector) with a single process, which has the potential to use local image context to determine a cell's type. This method is significantly more involved, more black boxy and more computationally expensive than the standard two-step process, so it should be definitively demonstrated that it is worth the trouble by showing that it performs better than the two-step process. Unfortunately, this does not seem to be the case, and the paper appears to hide this fact in a supplementary figure.

Most papers have two types of figures/analyses: the main analyses, which show that something works significantly better than prior art, and some secondary analyses, which show why it works better, how it can be applied, how performance varies with parameters etc. This paper does not have the main analyses, but it has secondary analyses and it does those really well. A lot of detail is put into carefully determining if the CNN indeed takes advantage of context to do its classification for example, which I think is convincingly proved. However, that analysis is only valuable if it can first be shown that the new proposed method works significantly better than the two-step process, at least in some cases. The only mention of this is somewhere in the middle of the paper, where it is said that "XGBoost (...) got high prediction accuracies, globally comparable to CellSighter (Supp Fig. 2A)". This should be the main figure, and unfortunately the only thing it proves is that CellSighter is not worth the trouble over the much simpler, much faster, and more interpretable two-step process.

I would suggest that the authors dig a lot deeper into making this comparison and try to understand why they couldn't beat the two-step XGBoost method. Their new approach is theoretically very interesting and promising, but something appears to be holding them back

from being able to beat XGBoost. It is shown in some of the secondary analyses that XGBoost does not do as well as CellSighter at incorporating context information (because it does not see any context), which would imply that CellSighter loses performances elsewhere. Where? Perhaps if that can be investigated and fixed, CellSighter would be able to consistently outperform XGBoost.

Assuming the authors can convincingly beat XGBoost, I would be happy to review a revised version of this paper. The revised paper should include comparisons with the two-step method (XGBoost) for each quantitative figure, and the result across cell classes should be summarized with single scores and compared between methods. I really do hope the authors can beat XGBoost, because the method is theoretically very interesting, despite being complex.

We thank the reviewer for this important comment regarding the comparison to XGBoost, as well as for the positive evaluation of our secondary analyses regarding context. We would like to address the reviewer's comments in two ways. First, we will explain why in the original manuscript the performance of CellSighter was allegedly comparable to XGBoost, and why we opted to focus on the context analyses. Next, we will provide new data and analyses demonstrating that, when evaluated correctly, CellSighter is in fact superior to XGBoost.

One of the biggest challenges in applying supervised learning for cell classification is obtaining good ground-truth data for training and evaluation. All datasets currently available in the public domain were generated by a combination of gating and clustering of the expression matrices. As such, it is very easy to train an ML-based model (e.g. XGBoost) that will just learn the clustering. This will give excellent results in a confusion analysis, but will not reflect any real benefit in accuracy. In the original paper we took a hybrid approach, which we believed would be a good compromise between labeling accuracy and throughput. We over-clustered 100K cells to 100 clusters, and annotated each of these clusters as one of 14 labels. While for some clusters the identity was clear from the heatmap, for many it was unclear due to spillover, and we had to visually inspect the images to assign this cluster to the label that was most probable for most cells in the cluster. **PbP Fig.1** shows an example for a difficult-to-assign cluster, in which the cells express both proteins of T cells (CD3) and proteins of B cells (CD20).

PbP Figure 1: Clustering expression vectors yields ambiguous cell classes. (A) The expression vectors for all cells in the dataset were clustered to 100 clusters using FlowSOM¹. Each cluster was then manually annotated. (B) A scatter plot of the cells from the clusters marked by the green, pink and blue boxes in A. For each cell, shown is its expression of CD3 (x-axis) and CD20 (y-axis). Cells from the green cluster can be classified as T cells, since they express CD3, but not CD20. Cells from the pink cluster can be classified as B cells, since they express CD20, but not CD3. Cells from the blue cluster are difficult to classify, and encompass T cells next to B cells, B cells next to T cells and ambiguous cells, which could plausibly be either T cells or B cells.

Consequently, while the *Expert Annotation* on which we trained and evaluated our models is a reasonable approximation of ground truth labels, it is by no means 100% accurate. For this reason, when we found that XGBoost and CellSighter yielded similar confusion matrices in the original manuscript, it was hard to evaluate whether this was because of lower performance in each one of the models, errors in expert annotation or both. We therefore turned to visual inspection of the images, and found that XGBoost had many cases of obvious misclassifications “that a human would not make”. This then drove us to focus on the context analysis in the original manuscript and show that CellSighter was superior in its ability to use this information for classification.

Following the reviewer’s comment, we now realize that focusing on context is not enough and that we require a comparison to a gold-standard benchmark that reflects ground-truth. For this reason, we performed several analyses:

1. **We manually labeled 900 cells** in the original four test images used for the study which differed in classification between the *Expert Annotation* and *CellSighter prediction* (off-diagonal cells in Figure 1D of the original manuscript). The annotation was performed blindly, by a different student, who received two suggestions for labels and had to decide whether label 1 was correct, label 2 was correct, both were plausible or none were correct. The order of the labels (Expert Annotation vs. CellSighter) was randomized to avoid any potential biases. We found that CellSighter was correct in ~41% of the cells, the expert annotation was correct in ~31% of the cells, both were plausible in ~16% of the cells and none were correct in ~11% of the cells (**PbP Fig. 2**). **Altogether, we find that the annotations of CellSighter are superior to what we previously used as our benchmark.** Gratifyingly, although CellSighter was trained on imperfect labels, generated by clustering, it was still able to generalize and learn spatial information to assist in its predictions.

PbP Figure 2: Manual analysis reveals CellSighter labels to be more accurate than ‘ground truth’ used for training. 900 cells from the test images, differing in classification between *CellSighter* and *Ground truth*, were subjected to manual inspection, to determine which classification was correct. Results show that CellSighter labels are superior to the original labels used for train and test.

- Following this analysis, we decided to test our approach on real manually-labeled data. To this end, we turned to another melanoma dataset of 111 $0.5 \times 0.5 \text{mm}^2$ images, encompassing 145,668 cells and generated highly-curated ground-truth labels for all cells using gating and sequential rounds of visual inspection and manual annotation. Importantly, we labeled comprehensive FOVs and did not allow the annotators to just pick cells for labeling, to avoid focusing on ‘easy’ cells. **To our knowledge, this data comprises the first comprehensively manually-annotated multiplexed imaging dataset**, and we foresee that it will serve as a valuable resource for the community.
- Next, we trained both CellSighter and XGBoost on 101 images, and predicted the cell labels for the remaining 10 images. We made sure that different images from the same patient were grouped as either train or test, but not both, to avoid overfitting on any specific patient.

We found that CellSighter outperformed XGBoost in eight out of ten categories and has the same performance on the remaining two categories, with an average increase in recall of 4% (recall $88.6 \pm 7\%$ compared with 84.5 ± 9 , **PbP Fig. 3**). Following the reviewer’s comment, we now present this analysis in figure 1D-F of the revised manuscript.

We thank the reviewer for their valuable input, as we think that this analysis greatly increases the strength of the paper.

PbP Figure 3: CellSighter outperforms XGBoost and clustering in cell classification. (A) For all cells in the test set, shown is the comparison between manually-curated labels (x-axis) and XGBoost predictions (y-axis). **(B)** Same as (A), for CellSighter predictions. **(C)** Shown is the recall for each cell class compared to manually-curated labels, for CellSighter (blue), XGBoost (orange) and Clustering (green). **(D)** Same as (C) for balanced accuracy.

There should also be a comparison of processing speed. CellSighter requires a neural network to be run for each and every cell in the image, unless I misunderstood. Since many datasets are in the many thousands or millions of cells, this can quickly lead to prohibitively large runtime costs.

We thank the reviewer for this important point, also raised by reviewer 3. Training with one GPU (Quadro RTX 6000), one epoch, which is equal to one pass over the whole train set, takes about 6 minutes for one model in the ensemble; training can be done in parallel. For the datasets in the manuscript, we trained for 10 to 40 epochs, depending on the dataset. So altogether, training on one GPU took 1-4 hours. Evaluation on new data takes approximately 40 seconds for 1000 cells, so ~7 minutes for 100,000 cells. Inference can easily be done in parallel to reduce running times. Naturally, the number of cells for training, crop size, batch size, RAM availability and other parameters can affect the running time. Altogether, in our experience run times were not prohibitive. In fact, these remain negligible compared to the time that it takes to perform the initial labeling using gating, clustering, visual inspection and manual annotation. Importantly, this time is invested by a computer in CellSighter, as opposed to time invested by human researchers to curate clustering results in current approaches. Following the reviewer’s comment, we have now added this information to both the results and methods sections of the manuscript (pages 5-6 and 18 respectively).

Reviewer #2 (Remarks to the Author):

Multiplexed tissue profiling constitutes one of the cutting-edge approaches to studying the tumor microenvironment as it provides spatial context to single-cell epitope co-localization and therefore cells' functional status. Although this is a promising technology, several technical problems and time-consuming barriers limit its application in clinical practice.

In this work, authors implement a deep-learning workflow, Cellsighter, to streamline cell classification from multiplex imaging to reduce labor-intensive work and to improve on current expression matrix-based cell classification. This neural network approach provides an effective and efficient cell classification with a limited training set, which seems to be less susceptible to some of the bias related to current methods, including spillover. In my view, this method is of great interest to the field as it could allow a wider application of multiplex technology.

The article and abstract are overall clear, well written, and detail the various experiments conducted by the authors. References are well-detailed. I did not identify any major flaws in the methodology and statistical results, which are in my view original, well-conducted, and reported. The conclusions provided are in line with the findings, robust, valid, and reliable, and do not exaggerate the benefits provided by this approach.

We thank the reviewer for the positive assessment of our work.

I'd like the author to specify two aspects:

- The authors tested this approach also on fluorescence data reporting a lower performance on "several classes", without detailing the problems. IF-based approaches are more diffuse than MIBI but are affected by several imaging problems which could greatly benefit from a neural-network assisted interpretation. I'd like, if possible, to more clearly detail the reason of the low performance and the limits in evaluating a fluorescence imaging with this CNN methods.

We thank the reviewer for this important question, also raised by reviewer 3. For the CODEX data, we used a publicly available dataset from Schurch et al., and trained CellSighter on labels provided by the authors. Following the reviewer's question, we went back to the cell annotations and identified some inaccuracies. Critically, this dataset was published in 2020, before the introduction of AI-based segmentation algorithms, and as such it contained many segmentation errors, as highlighted in PbP Figure 4A. We therefore used Mesmer² to resegment the cells, and generated highly-curated ground-truth labels for 85,179 cells, from 35 FOVs using gating and sequential rounds of visual inspection and manual annotation. PbP Figure 4B-C shows the influence of this process on cell classification in two exemplary FOVs.

PbP Figure 4: Improvements in cell segmentation and classification of data from Schurch et al. ³ **(A)** Data from Schurch et al. was segmented using Mesmer ² to improve segmentation. **(B)** For two FOVs, shown are the results of cell classification from the manuscript (left column) and following our segmentation and annotation (right column). **(C)** For the two FOVs in (B), shown are the changes in the number of cells of different types in Schurch et al. (magenta) and after resegmentation and expert annotation (green).

We used these novel labels to train CellSighter on 66,691 cells from 27 FOVs, and evaluated on the remaining 8 FOVs. We found that the results were dramatically improved (recall 80 ± 9 compared to 70 ± 19), as shown in PbP Figure 5. Following the reviewer's comment, we have now replaced the fluorescence analysis in the paper with this analysis, which better demonstrates CellSighter's performance on fluorescent data (**Fig. 5A,B**). We thank the reviewer for this important contribution to improving our paper.

PbP Figure 5: CellSighter accurately classifies fluorescent images. Images from Schurch et al. ³ were segmented using Mesmer ² and cells were classified into thirteen classes. For all cells in the test set, shown is the comparison between manually-curated labels (x-axis) and CellSighter predictions (y-axis).

- They mention that the CNN model was trained on 10 separate-trained CNN, which should refer in my understanding to 10 cell lines. In methods I find a referral to 14 cells lines (B cells, DC, CD4, Treg, Macrophages, Cd8, Stroma, Germinal B cells, HEV, CD4 memory, NK, Neutrophils, Endothelial, Tumor). If that's the case, I'd like this aspect to be more clearly stated.

We apologize for the lack of clarity. CellSighter performs multi-class classification. As such, one CNN is used for all cell types (i.e., 14 in the case of the melanoma lymph node dataset). The CNN receives a small crop, centered around a cell of interest, and outputs one of 14 labels, e.g. "B cell", "CD4 T cell" etc. Altogether we train an ensemble of ten such models. We then set the label as the majority vote of these ten models, and set the confidence to be the average confidence of all ten models. We took this approach since we found that the predictions made by the ensemble are more robust than the predictions made by any individual model and the confidence is beneficial for filtering the results. This property of ensembles has been previously demonstrated in the machine-learning and deep learning literature ⁴. Following the reviewer's comment, we now clarify and elaborate on this point in the main text (page 4) and methods (page 17-18). We have also revised the schematic illustration of the model in figure 1C to better reflect this architecture.

Reviewer #3 (Remarks to the Author):

The manuscript by Amitay et al. describes a novel approach, termed CellSighter, that employs neural networks to classify cell types in multiplexed imaging data. The authors demonstrate that CellSighter reproduces cell type labels obtained with currently employed methods, that these classifications are based on protein co-expression as well as the subcellular location of the proteins and they show how manipulating features of this framework impacts predictions. Lastly, they show that CellSighter can be applied to datasets from various technologies.

I believe that the manuscript constitutes an important advance in the field of spatial biology and multiplexed imaging and their approach to incorporate the imaging data for the task of cell classification is novel. Most of the claims in the paper are supported well, with the exception of my comments below. The work seems to be of high quality overall and appropriate methodologies were used.

We thank the reviewer for the positive assessment of our work.

Comments:

The manuscript mentions in several places that CellSighter would accelerate or speed up the process of cell classification. I am not sure this is true, and I don't think this is shown. In my understanding, to use CellSighter one still has to provide training data for which one either has to manually identify cells or alternatively go through the standard pipeline of segmentation and clustering. So overall, these time-consuming steps still seem part of the pipeline. The authors suggest that "It is easier and faster to classify thousands of cells than to classify millions of cells." but I am not fully convinced. Of course, clustering algorithm runtimes go up a bit, but many widely used algorithms can process millions of cells within minutes. Even in "manual gating" one often looks at all cells at once and cell numbers do not drastically influence the time it takes to identify cell populations. I suggest the authors remove this claim from the manuscript or show a convincing analysis of how their proposed workflow saves time.

We thank the reviewer for this important comment since we realize now that the process for generating the ground-truth labels was not described clearly enough in the original manuscript. The reviewer is correct that the time that it takes to run clustering is not prohibitive for some algorithms (albeit often traded for accuracy), and in any case can be comparable to the time that it takes to train a CNN. Similarly, the reviewer is correct that manual gating can be performed on 10K or 100K cells in a similar amount of time. The problem is that when working on cells segmented from images, the data is very noisy due to errors in segmentation, spillover of signal from neighboring cells, treating a 3D tissue as 2D and more. As a result, both clustering and gating fall short in their accuracy. A good example of this is illustrated in **PbP Figure 6** below. Panel A depicts the results of clustering the cells to 100 clusters using FlowSOM ¹. Several clusters are easy to classify biologically. For example, the clusters highlighted with a pink rectangle that have expression of CD20 and CD45RA and represent B cells; or the clusters highlighted with a green rectangle that have expression of CD3 and CD45RO and represent T cells. However, there is a large subset of clusters in which the cells have coexpression of CD20 and CD3 (cyan rectangle). Prior biological knowledge informs us that CD20 and CD3 are usually not coexpressed on the

same cell. As such, these cells are a mixture of B cells with spillover from adjacent T cells, or T cells with spillover from adjacent B cells or ambiguous cells in which imaging or segmentation artifacts result in the appearance of coexpression of CD3 and CD20 (**PbP Fig. 6B**). It is impossible to discern between these options based on clustering alone. Generating accurate classifications for these cells requires inspection of images and manual annotation (right panel). For this reason, classification of multiplexed images never relies solely on clustering and will ultimately include a manual inspection and annotation step.

This second, largely-manual part of the process is the part that we describe to be highly time-consuming and elaborate. Naturally, this part scales with the number of ambiguous cells, which scales with the size of the dataset. In our experience, this process often takes several weeks, depending on the diligence of the student. For this reason, we repeatedly state that “It is easier and faster to classify thousands of cells than to classify millions of cells”. Following the reviewer’s comment, we now understand that this arduous process was not conveyed clearly enough in the original manuscript. We therefore now revised the text to better detail the process in the introduction (page 3) and in the results (page 5-6). We also added the figure below to the manuscript as supplementary figure S1A. We thank the reviewer for helping us clarify this important point.

PbP Figure 6: Clustering expression vectors yields ambiguous cell classes. (A) The expression vectors for all cells in the dataset were clustered to 100 clusters using FlowSOM¹. Each cluster was then manually annotated. **(B)** A scatter plot of the cells from the clusters marked by the green, pink and blue boxes in A. For each cell, shown is its expression of CD3 (x-axis) and CD20 (y-axis). Cells from the green cluster can be classified as T cells, since they express CD3, but not CD20. Cells from the pink cluster can be classified as B cells, since they express CD20, but not CD3. Cells from the blue cluster are difficult to classify, and encompass T cells next to B cells, B cells next to T cells and ambiguous cells, which could plausibly be either T cells or B cells.

The analysis shown in Figure 1 indicates that CellSighter accurately reproduces cell labels that were generated in more established ways, including through iterative clustering. As expected, there is some disagreement between the two. However, I am surprised that for these cells with disagreeing labels, it was not clear that CellSighter had produced more accurate labels. Does this imply that FlowSOM clustering, which I believed was used here, works equally well as CellSighter?

I find this to be at odds with the later analysis showing superior performance of CellSighter compared to a machine learning classification approach. How do the authors explain this?

We believe that this comment is closely related to the previous one. As we explain in our answer to the previous comment, the expert annotation is not merely clustering. It is indeed based on clustering, but the results of the clustering further underwent significant manual curation by a human, which took several weeks to complete.

To address the reviewer's question, we now use our ground truth, manually-curated labels for the melanoma dataset and perform a head-to-head comparison of the performance of CellSighter to clustering without extensive manual curation. To perform clustering, we employed a widely-used approach, which is to over-cluster the data to 100 clusters and then annotate each of these clusters to fewer biologically-relevant clusters based on the expression matrix. Notably, some of the manually-curated labels, such as Tregs, could not be identified by this clustering at all, presumably due to their low abundance in the data, and we needed to gate them separately. For ambiguous clusters, we chose the most probable label according to the expression matrix. We then benchmarked CellSighter and clustering according to manually-labeled data and found that CellSighter drastically outperformed clustering across all classes (recall 88.6 ± 7 compared to 75 ± 16 , **PbP Figure 7**).

Following the reviewer's comments, we have now significantly modified figure 1 of our manuscript. We better explain the process of generating ground-truth labels and show a direct comparison between CellSighter and conventional clustering. We thank the reviewer for raising this point and improving our manuscript.

PbP Figure 7: CellSighter outperforms clustering in cell classification. Shown is the recall (A) and balanced accuracy (B) for each cell class compared to manually-curated labels, for CellSighter (blue) and Clustering (orange).

On the technical side, in many clustering pipelines it is common that integrated single-cell protein expression levels are normalized (e.g. asinh or log) and scaled. Was this done before the imaging data was fed into CellSighter? Furthermore, what would be the effect of channel-to-channel normalization, e.g. using the 99th percentile? Could this be used to improve performance?

Our current workflow for CellSighter does not normalize the images. Following the reviewer’s comments, we tested the effects of the following normalization scheme, as suggested by the reviewer:

1. Anscombe transform. This transformation has a similar effect to asinh or log in that it reduces the effect of the heavy tail in the distribution, but it is more commonly used for low-count images such as the ones used in this paper.
2. Scale pixels according to the 99th percentile.

We find that, compared to using non-normalized data, anscombe transformation reduced the model’s performance (recall 77 ± 15 , **PbP Fig. 8**). Normalizing the pixels to the 99th percentile yields comparable results to not performing any normalization (recall 86 ± 9 compared to 85 ± 8). Interestingly, some classes improve (e.g. NK cells), whereas others are reduced (e.g. Macrophages). While we did not investigate this thoroughly, we speculate that the dynamic range of the intensity of the pixels holds valuable information, that may be eroded when normalizing. These results should naturally be explored in additional datasets. Following the reviewer’s comments, we now added this analysis of normalization options to the manuscript as supplementary figure S3 D,E, and discuss it in the results section on page 12 and 21.

PbP Figure 8: CellSighter performance using different kinds of normalizations. For all cells in the test set, shown is the comparison between manually-curated labels (x-axis) and CellSighter predictions (y-axis) for either non-normalized data (left), data normalized to the 99th percentile (middle) and anscombe-transformed data (right).

I think it would be important to show how the number of clusters (which still has to be chosen manually) affects lineage identification accuracy. Is there any value in “over-clustering” the data followed by pooling of some clusters to arrive at biologically meaningful populations? Or would this reduce overall accuracy?

We thank the reviewer for this important question, also raised by reviewer 2. For the CODEX data, we used a publicly available dataset from Schurch et al., and trained CellSighter on labels provided by the authors. Following the reviewers' questions, we went back to the cell annotations and identified some inaccuracies. Critically, this dataset was published in 2020, before the introduction of AI-based segmentation algorithms, and as such it contained a non-negligible amount of segmentation errors, as highlighted in PbP Figure 10A. We therefore used Mesmer² to resegment the cells, and generated highly-curated ground-truth labels for 85,179 cells, from 35 FOVs using gating and sequential rounds of visual inspection and manual annotation. PbP Figure 10B-C shows the influence of this process on cell classification in two exemplary FOVs.

PbP Figure 10: Improvements in cell segmentation and classification of data from Schurch et al.³ (A) Data from Schurch et al. was segmented using Mesmer² to improve segmentation. (B) For two FOVs, shown are the results of cell classification from the manuscript (left column) and following our segmentation and annotation (right column). (C) For the two FOVs in (B), shown are the changes in the number of cells of different types in Schurch et al. (magenta) and after resegmentation and expert annotation (green).

We used these novel labels to train CellSighter on 66,691 cells from 27 FOVs, and evaluated on the remaining 8 FOVs. We found that the results were dramatically improved (recall 80 ± 9 compared to 70 ± 19), as shown in PbP Figure 11. Following the reviewer's comment, we have now replaced the fluorescence analysis in the paper with this analysis, which better

demonstrates CellSighter’s performance on fluorescent data (**Fig. 5A,B**). We thank the reviewer for this important contribution to improving our paper.

PbP Figure 11: CellSighter accurately classifies fluorescent images. Images from Schurch et al. ³ were segmented using DeepCell and cells were classified into thirteen classes. For all cells in the test set, shown is the comparison between manually-curated labels (x-axis) and CellSighter predictions (y-axis).

Related, it would be much appreciated to show how imaging resolution affects cell classification, given the differences in resolution provided by different approaches to multiplexed imaging.

We thank the reviewer for this interesting suggestion. Following the reviewer’s comment, we have now tested CellSighter’s performance while varying the resolution. To allow a head-to-head comparison, we used the same dataset at the original resolution (0.5µm/pixel) and then simulated from this data images at a resolution of 1µm/pixel using a 2x2 kernel. We found that when reducing the resolution, the results were largely comparable (**PbP Fig. 12**, recall 85±7 compared to 85±8). Importantly, reducing the resolution reduced the time to train the model by 166% (from 5 minutes per epoch to 3 minutes per epoch). From this analysis we conclude that CellSighter can work equally well on images taken at a resolution of 1 µm/pixel and it trains faster. Resolution is an important parameter in determining the time that it takes to acquire multiplexed images for many imaging modalities. As such, these comparable results are valuable for the community since they suggest that for the task of cell classification a resolution of 0.5µm/pixel is not required, and could be traded for a reduction in imaging and analysis runtimes. We have now

added this analysis to the manuscript in the results (page 12), methods (page 21) and supplementary figure S3 G.

PbP Figure 12: CellSighter performs comparably at resolutions of 1µm/pixel. For all cells in the test set, shown is the comparison between manually-curated labels (x-axis) and CellSighter predictions (y-axis) for the data at a resolution of 0.5µm/pixel (left) or after applying a 2x2 kernel to artificially reduce the resolution to 1µm/pixel (right).

Given the issues with technical and biological factors as laid out in the manuscript: Is there a way to express prediction confidence on the pixel level? While not critical, I think this would be a huge improvement in the field to give users a way to disregard such overlapping or otherwise confounding pixels from the subsequent analysis.

This is a very interesting idea, which we actually haven't considered. CellSighter is a cell-based classifier, and thus outputs labels and confidence at the level of the cell. However, the CNN indeed contains confidence information at the pixel level. We can see this when we examine the gradients of the network, which indicate which pixels in which channels are contributing most strongly to the prediction. For example, in PbP Fig. 13 below we see a T regulatory cell and the pixels with the highest gradients, which are dominant in influencing the prediction, are the center-most pixels, which display positivity for FoxP3. It is possible to think about how to incorporate this information into a pixel-level confidence score, but that would require some more work and thought. It is also unclear that in this instance a cell-based classifier would be the method of choice, and one may prefer pixel-level dimensionality reduction, such as using autoencoders as suggested by others⁵. Altogether, while we believe that it is an interesting idea, it is also outside the scope of this paper. Following the reviewer's comment, we now discuss such avenues for future work in the discussion (page 15).

PbP Figure 13: CellSighter performs comparably at resolutions of 1µm/pixel. Shown are expression levels (bottom) and normalized gradients of the CNN (top) for a single cell classified by CellSighter as a Treg, in the images of FoxP3 (left), CD4 (middle) and CD45 (right). The images of the gradients show which proteins most influence classification and where the network is looking for them.

It seems that the presented analysis was performed using commonly employed “lineage-markers” that mostly are expressed on one or few cell populations. Given that these technologies are able to quantify about 40-50 proteins, is there any value in including more proteins for the CellSighter analysis or can this also negatively impact classification performance?

We thank the reviewer for this important point. In common practices of analysis of multiplexed imaging, it is common to first assign cell types using lineage markers and then evaluate the expression of functional proteins on these cells⁶⁻⁸. The logic behind this workflow is that ultimately multiplexed images don’t have that many proteins (indeed 40-50) and often, a single protein can influence classification (eg, Tregs are most often separated from CD4 T cells only by FoxP3). As such, one doesn’t want the state of the cell to influence classification quality. For example, if in one’s dataset tumor cells tend to proliferate more than other cells and express more Ki-67, one doesn’t want proliferation and Ki-67 to be naturally connected to tumor cell identity, in order to not misclassify proliferating B cells, for example. For this reason, our approach conforms to existing practices and uses only the lineage proteins to classify cell types.

It is indeed possible, as suggested by the reviewer, to also use functional markers. To evaluate the effect of this on classification accuracy, we retrained CellSighter on the same dataset using either 25 lineage proteins, or 39 lineage+functional proteins. We found that the results were largely comparable (**PbP Fig. 14**, recall 85 ± 8 compared to 85 ± 8). As such, we conclude that adding phenotypic proteins does not drastically influence the accuracy of classifying cell types for this particular combination of dataset and proteins. This could of course change depending on dataset and proteins visualized. Following the reviewer’s comment, we now discuss this analysis in the manuscript on page 12 and 21, supp. Figure 3F.

PbP Figure 14: CellSighter performs comparably when trained with either lineage proteins or lineage and functional proteins. For all cells in the test set, shown is the comparison between manually-curated labels (x-axis) and CellSighter predictions (y-axis) for the data trained with 25 lineage proteins (left) or 39 lineage and functional proteins (right).

Minor comments:

Not being an expert on CNNs, I was not familiar with the specifics of the term gradient. I think the concept should be introduced shortly in the manuscript, given the broad readership of Nature Communications.

Following the reviewer's comment, we now more clearly explain the term gradient as it rates to neural networks. See Results section page 7 and Methods section page 19.

Fig 2B what is the color scheme showing?

We apologize for the confusion. It seems that the title of the colorbar was accidentally removed when preparing the pdf. The color scheme shows the correlation between the expression of any particular protein (x-axis) and the confidence of CellSighter in predicting any particular cell type (y-axis). Since these are correlation values, they range from -1 (red) when the expression of the protein is anticorrelated with the class to 1 (green) when the expression of the protein is correlated with the class.

References:

1. Van Gassen, S. *et al.* FlowSOM: Using self-organizing maps for visualization and interpretation of cytometry data. *Cytom. Part A* **87**, 636–645 (2015).
2. Greenwald, N. F. *et al.* Whole-cell segmentation of tissue images with human-level performance using large-scale data annotation and deep learning. *Nat. Biotechnol.* (2021) doi:10.1038/S41587-021-01094-0.
3. Schürch, C. M. *et al.* Coordinated Cellular Neighborhoods Orchestrate Antitumoral Immunity at the Colorectal Cancer Invasive Front. *Cell* **182**, 1341-1359.e19 (2020).
4. Dietterich, T. G. Ensemble Methods in Machine Learning. *Lect. Notes Comput. Sci. (including Subser. Lect. Notes Artif. Intell. Lect. Notes Bioinformatics)* **1857 LNCS**, 1–15 (2000).
5. Spitzer, H., Berry, S., Donoghoe, M., Pelkmans, L. & Theis, F. J. Learning consistent subcellular landmarks to quantify changes in multiplexed protein maps. *bioRxiv* 2022.05.07.490900 (2022) doi:10.1101/2022.05.07.490900.
6. Keren, L. *et al.* A Structured Tumor-Immune Microenvironment in Triple Negative Breast Cancer Revealed by Multiplexed Ion Beam Imaging. *Cell* **174**, (2018).
7. McCaffrey, E. F. *et al.* The immunoregulatory landscape of human tuberculosis granulomas. *Nat. Immunol.* 2022 1–12 (2022) doi:10.1038/s41590-021-01121-x.
8. Hoch, T. *et al.* Multiplexed imaging mass cytometry of the chemokine milieu in melanoma characterizes features of the response to immunotherapy. *Sci. Immunol.* **7**, (2022).

REVIEWERS' COMMENTS

Reviewer #1 (Remarks to the Author):

The authors have addressed my concerns in a satisfactory manner (thanks) and I don't have any follow up concerns.

Reviewer #2 (Remarks to the Author):

The authors responded to my questions and clarifications appropriately. The manuscript should, in my view, be considered for publication.

Reviewer #3 (Remarks to the Author):

The authors have responded appropriately to all my previous comments. I am happy to see that these new analyses and clarifications have improved that manuscript which I think will be a great tool the the multiplexed imaging community.